# A hydro-topological strategy enables self-regulating biofilms for sustainable wastewater treatment

Yong Fang [1], Zhiqiang Zhang [2], Boru Xue [3], Ying Liu [1] ✉ & Kuichuan Sheng [1] ✉

The moving bed biofilm reactor (MBBR) is a cornerstone technology in modern wastewater treatment, yet its performance is often hindered by carrier clogging, which significantly reduces overall treatment efficiency and undermines the technology's environmental benefits. Here, we introduce a V-carrier hydro-topological design strategy that enables biofilm self-regulation, allowing simultaneous control of biofilm thickness and continuous hydraulic shear-induced self-cleaning. In a pure biofilm system treating real municipal wastewater for over 500 days, the V-carrier achieves stable and efficient nutrient removal even under low-temperature (9.1 °C) and high-loading conditions. Crucially, it achieves a 3.2-fold higher unit biomass nitrification rate with a biofilm biomass 44% lower than a conventional K3 carrier, demonstrating that treatment efficiency is decoupled from biomass quantity through optimal ecological niche design. This work establishes a paradigm for biofilm reactor design, transforming carriers from passive substrates into active regulators of microbial ecosystems, with profound implications for sustainable water infrastructure.

Wastewater treatment plants (WWTPs) are critical infrastructure for protecting public health and environmental quality, directly supporting 11 of the 17 Sustainable Development Goals (SDGs)[1–4]. Among advanced treatment technologies, the moving bed biofilm reactor (MBBR) process supports SDGs 6, 11, and 12 through its ability, enabled by suspended biofilm carriers, to deliver high nutrient removal efficiency (SDG 6), a compact physical footprint (SDG 11), and reduced sludge production (SDG 12). Through these contributions to water security, urban sustainability, and resource efficiency, MBBR has been established as a leading technology for new and upgraded WWTPs worldwide[5–7].

Over four decades of development have yielded numerous MBBR carrier designs for engineering applications[8,9]. These efforts, even using advanced tools like 3D printing techniques, have predominantly pursued one goal: maximizing the protected surface area to enhance biofilm attachment[10]. However, increased surface area has not consistently improved treatment efficiency[11], revealing a deeper issue rooted in two factors: a paradigm inherited from suspended-growth systems (e.g., activated sludge), where more biomass implies better performance, and a widespread lack of long-term operational data. This has led to the misapplication of the more biomass equals more efficiency principle to biofilms—systems where controlled biofilm thickness and metabolic balance are paramount. The design focus on maximizing protected surface area has promoted uncontrolled biofilm growth, with carrier clogging even misinterpreted as a positive sign[12,13], thereby diverting attention from the need for biofilm managed growth. This fundamental misdirection manifests as the core limitations of conventional MBBR systems: severe carrier clogging and scaling, drastically diminishing the effective surface area, impairing mass transfer, and disrupting microbial community structure by fostering competition for oxygen and substrates among unfavorable species[14–16]. Moreover, clogged carriers exhibit increased buoyant

[1]College of Biosystems Engineering and Food Science, Zhejiang University, Hangzhou, China. [2]Hangzhou Tao of Water Technology Co., Ltd, Hangzhou, China. [3]School of Mining Engineering, University of Science and Technology Liaoning, Anshan, China. ✉e-mail: liuyingzju@zju.edu.cn; kcsheng@zju.edu.cn

density, demanding higher energy inputs for mixing and suspension, thereby elevating operational costs and environmental impacts. Consequently, the sustainability benefits of conventional MBBR systems are compromised by these inherent design flaws, presenting a significant challenge to achieving the SDG targets they purport to advance.

The biofilm thickness is a well-established critical parameter, governing the structure of the microbial community, assembly processes, mass transfer dynamics, and the potential for micropollutant removal[17–20]. Despite its recognized importance, a fundamental practical limitation remains: no existing MBBR carrier can reliably control biofilm thickness under the dynamic hydraulic conditions of full-scale reactors[21,22]. One notable effort to address this limitation was the Z-carrier (AnoxKaldnes), which was based on the geometric premise that its grid height would regulate biofilm thickness[17,23,24]—a design that also had implications for mitigating clogging and scaling due to its open structure. However, their own long-term evaluation revealed a critical flaw: biofilms at the center of the grid cells became significantly thinner or were entirely absent[25]. This outcome underscores a fundamental stalemate in sustainable carrier design, revealing a poor understanding of the hydrodynamic mechanisms that control biofilm development. The failure of this simplistic geometric approach highlights the necessity for a design paradigm rooted in hydraulic principles.

The propensity for clogging in conventional tubular carriers is intrinsically linked to their hydraulic design. Carriers with large internal diameters (e.g., K3) demonstrate clogging resistance, whereas those with small hollows (e.g., K5, Biofilm-Chip P, Biofilm-Chip M) are prone to complete clogging during long-term operation[15]. This phenomenon is explained by the Hagen-Poiseuille equation, which dictates that a smaller tube diameter, under identical hydraulic conditions, results in a greater pressure drop and fluid resistance. This, in turn, leads to a lower internal flow velocity and shear stress[26–28]. We therefore hypothesize that clogging in small-tube carriers results from

insufficient hydraulic shear to balance biofilm growth with detachment, allowing uncontrolled accumulation. This principle is exemplified by fixed-bed filters, which maintain biofilm thickness and mass transfer efficiency through periodic backwashing—a process that generates high transient shear to scour excess biomass[29–31]. In an MBBR, however, the maximum continuous hydraulic shear is exerted only on the biofilm surface directly exposed to the bulk liquid flow. This critical observation suggests that the strategic orientation of the biofilm surface is paramount for harnessing hydraulic forces for continuous self-cleaning.

Building on this foundation, we propose a hydro-topological strategy for autonomous biofilm regulation. This approach engineers the carrier's topology to control both biofilm thickness and its spatial orientation, thereby enabling direct and continuous hydraulic shearing of the outer surface to maintain a stable biofilm with a controlled, optimal thickness and high metabolic activity. We materialized this strategy by designing a V-carrier (Fig. 1), which operates on four synergistic principles:

1. Directed Growth: The V-shaped opening guides biofilm development outward, toward the liquid phase, ensuring efficient nutrient access for outer-layer cells and real-time clearance of metabolic products.
2. Thickness Constraint: The finite depth of the V-groove physically constrains the maximum biofilm thickness, intrinsically preventing the mass transfer limitations associated with overgrowth.
3. Biomass Protection: The inclined surfaces of the V-groove shield the established biofilm from detachment caused by excessive hydraulic shear forces.
4. Niche Stabilization: The continuous, staggered arrangement of V-grooves creates independent, consistent micro-chambers that function as minimal functional ecological niches, enhancing the overall stability and resilience of the biofilm community.

We hypothesize that this integrated V-carrier design enables stable, self-regulated biofilm maintenance, ensuring efficient long-

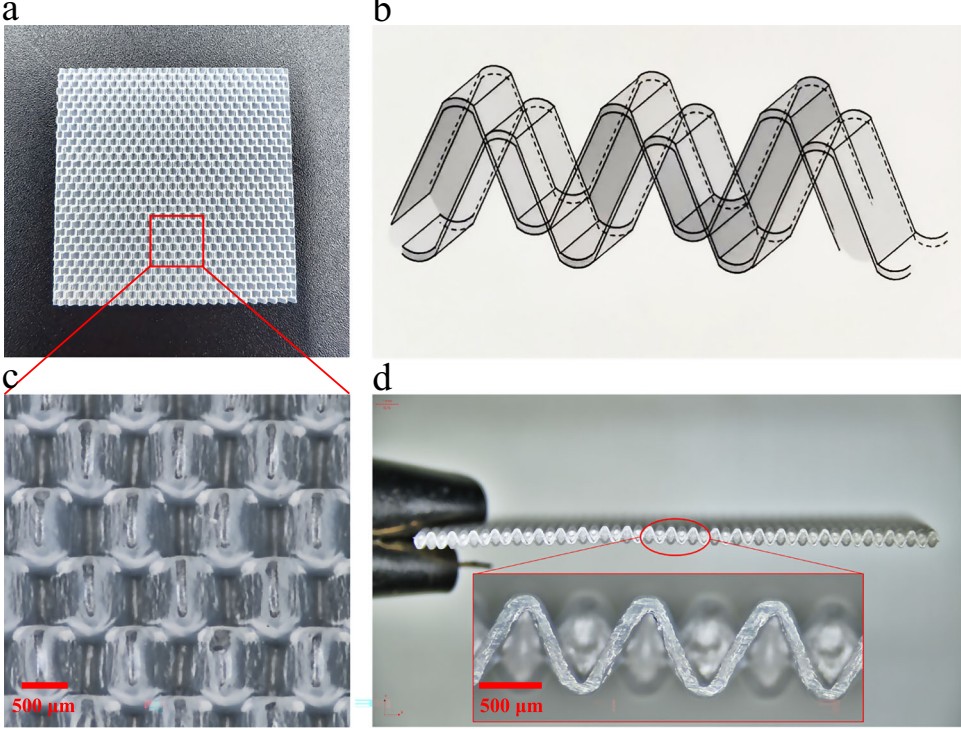

**Fig. 1 | Structure and key features of the V-carrier. a** Top view showing the square-sheet morphology (24 × 24 × 0.95 mm). **b** Schematic diagram illustrating the mirror-symmetric V-groove array (Panel created using Microsoft PowerPoint 2021). **c** Magnified top view revealing the individual V-grooves (scale bar: 500 μm). **d** Cross-sectional view demonstrating groove depth (0.72 mm) and oblique geometry (scale bars: 500 μm).

term mass transfer and treatment performance. To test this hypothesis, we operated a laboratory-scale anoxic/aerobic (A/O)-MBBR system treating real municipal wastewater for over 500 days, evaluating its potential as a sustainable solution for modern WWTPs.

## Results

### Design and characteristics of the V-carrier
The V-carrier used in this study was fabricated from high-density polyethylene (density 0.94–0.97 g cm$^{-3}$) to achieve near-neutral buoyancy for unrestricted movement in wastewater. As shown in Fig. 1a, the carrier exhibits a square-sheet morphology with overall dimensions of $24 \times 24 \times 0.95$ mm. Its core design consists of mirror-symmetric, staggered V-shaped grooves that serve as the primary attachment sites for biofilms (Fig. 1b). The groove depth is 0.72 mm, a key parameter for constraining the maximum biofilm thickness, while the opening length is 1.0 mm, yielding an opening ratio (length/depth) of 1.39 to optimize hydraulic exposure (Fig. 1c, d and Fig. 6a–f). The framework width is 0.7 mm, enabling each V-groove to function as an independent, consistent ecological niche. Each carrier contains 1536 such grooves, providing a total protected surface area of 1885 m$^2$ m$^{-3}$. The effective biofilm surface area, calculated from the outer surface of the biofilm, is a more functionally relevant 1,152 m$^2$ m$^{-3}$ for mass transfer calculations. The carrier's bulk density and structural porosity are $95 \pm 0.5$ kg m$^{-3}$ and 92%, respectively. These structural and material properties form the physical basis for testing the proposed hydro-topological strategy.

### Stable and efficient nutrient removal over long-term operation
The A/O MBBR system, packed exclusively with V-carriers, was operated for over 500 days while treating real municipal wastewater (Supplementary Fig. 1). Sodium acetate was supplemented to provide sufficient electron donors for denitrification, resulting in an operational C/N ratio of $4.9 \pm 0.3$ (Supplementary Fig. 2). The operational timeline was strategically divided into three phases based on hydraulic retention time (HRT) and temperature: start-up (Phase I, days 1–10), suitable temperature (Phase II-i, days 11–280; Phase II-ii, days 325–512), and low temperature (Phase III, days 281–324) (Fig. 2a). During the start-up phase (Phase I) at $15.6 \pm 0.9$ °C, the HRT was successfully reduced from 24 to 8 h within 10 days, demonstrating rapid biofilm establishment and system acclimation. In the subsequent suitable-temperature phase (Phase II-i, HRT = 8 h), the system achieved efficient nutrient removal, with effluent NH$_4^+$-N and total inorganic nitrogen (TIN) concentrations of $0.9 \pm 0.6$ mg L$^{-1}$ and $6.6 \pm 2.3$ mg L$^{-1}$, corresponding to removal efficiencies of $97.8 \pm 1.4$% (ammonia removal efficiency, ARE) and $86.7 \pm 5.2$% (TIN removal efficiency, NRE), respectively (Fig. 2b, c). When reactor temperature decreased to $10.4 \pm 1.2$ °C during Phase III, the HRT was increased to 9 h to compensate for reduced microbial activity. Despite this thermal challenge, the system maintained robust performance, with ARE and NRE of $97.0 \pm 0.5$% and $85.0 \pm 3.3$%, respectively. Upon temperature recovery (>14 °C) in Phase II-ii, the HRT was returned to 8 hours, resulting in further improved effluent quality (NH$_4^+$-N $= 0.5 \pm 0.4$ mg L$^{-1}$; TIN $= 4.6 \pm 1.7$ mg L$^{-1}$) and high, stable removal efficiencies (ARE $= 98.9 \pm 0.9$%; NRE $= 90.8 \pm 3.5$%). Throughout the entire operational period, the system exhibited high nutrient removal rates of $0.138 \pm 0.022$ g N g VSS$^{-1}$ day$^{-1}$ for ammonia (ammonia removal rate, ARR) and $0.587 \pm 0.108$ g N g VSS$^{-1}$ day$^{-1}$ for TIN (TIN removal rate, NRR) (Fig. 2d). Notably, the nutrient removal rates remained stable even during low-temperature operation (Supplementary Fig. 3), demonstrating minimal functional impairment and consistent with effective microbial adaptation. In line with the inherent characteristics of pure biofilm systems featuring extended sludge retention, phosphorus removal efficiency was variable (8–52%) (Supplementary Fig. 4), consistent with the limited biological phosphorus removal capacity under these conditions[32–34]. In practical applications, this limitation is

routinely and effectively overcome by coupling the biofilm process with a downstream chemical phosphorus removal step[35], which is a standard and cost-comparable practice also employed following conventional activated sludge processes to meet stringent effluent standards. Collectively, these results demonstrate that the V-carrier system delivers high and stable nutrient removal performance under varying hydraulic and temperature conditions, confirming its operational robustness.

### Mechanism of biofilm regulation—hydro-topological strategy
To elucidate the mechanisms enabling stable biofilm management, we conducted a comparative analysis of the V-carrier against conventional tube-carriers (Mutagbiochip, K5, and K3; Supplementary Fig. 5). The V-carrier maintained a stable volatile suspended solids-to-suspended solids (VSS/SS) ratio and consistent biofilm thickness of 500–550 µm—a range proven to benefit mass transfer and promote the removal of diverse micropollutants[17]—under both anoxic and aerobic conditions throughout long-term operation, demonstrating effective self-cleaning coupled with thickness control (Fig. 3a, b and Supplementary Note 1). Analysis of the Mutagbiochip carrier, designed for a -550 µm biofilm thickness on each side (Supplementary Fig. 6), revealed a critical limitation. Although it achieved the target thickness within 23 days—comparable to the optimal value for mass transfer[17,36]—significant inorganic accumulation (ash content increased from 18% to 56%; paired $t$-test, $P = 2.56 \times 10^{-6}$) and a substantial decline in the ARR occurred over 285 days (Fig. 3c and Supplementary Table 1). This demonstrates that geometric constraint for thickness control is insufficient without an integrated self-cleaning capability.

We posited that the V-carrier's open structure, which guides biofilm growth outward into the liquid phase (Fig. 3d-I), is key. In contrast, biofilms grow inward in tube-carriers (Fig. 3d-II). We hypothesize that sustained hydraulic shear on the exposed biofilm surface enables a dynamic balance between growth and decay, facilitating real-time metabolite clearance (Fig. 3e). Support for this comes from the K5 carrier (2 mm hollows), which clogged completely after 70 days, concomitant with a loss of hydraulic shear and a significant decrease in ARR from $0.032 \pm 0.003$ to $0.010 \pm 0.001$ g N g VSS$^{-1}$ day$^{-1}$ (paired $t$-test, $P = 2.53 \times 10^{-6}$; Fig. 3d, f). Conversely, the K3 carrier (5 mm hollows) maintained smooth inner-surface biofilms, a stable VSS/SS ratio ($0.70 \pm 0.02$), and a consistent thickness ($334 \pm 93$ µm) over 520 days, confirming the role of preserved hydraulic shear in sustaining mass transfer (Fig. 3d–II, g). To directly test the necessity of hydraulic exposure, we engineered a modified K5 (MK5) carrier with both open and hollow sections (Supplementary Fig. 5). While the open sites maintained a stable biofilm (VSS/SS $= 0.72 \pm 0.02$) for 520 days (Fig. 3d-III), the shielded hollows accumulated 57% inorganic precipitates (Fig. 3f). This controlled experiment provides definitive evidence that direct hydraulic shear is prerequisite for effective self-cleaning.

Finally, moving beyond simple geometric constraints, we asked whether topology alone could dictate biofilm thickness under realistic hydraulics. The MK5 carrier revealed a critical limitation: at its open sites, excessive hydraulic shear dominated over topological guidance, preventing biofilms from reaching the ridge height despite available space (Fig. 3d-III). We then tested the converse scenario using a U-carrier with reduced ridge spacing designed to attenuate shear. Here, the lack of hydraulic control led to spatial competition: biofilms preferentially colonized the shielded ridge surfaces, eventually bridging the U-grooves and causing clogging (Fig. 3h). Thus, neither unmanaged high shear nor excessive shielding succeeds; both fail to direct biofilm growth appropriately. The V-carrier succeeds by resolving this spatial competition through its hydro-topological design. The oblique grooves guide biofilm development into the liquid flow, ensuring that hydraulic shear, rather than overwhelming or being absent, functions collaboratively with topology to maintain an optimal, stable biofilm. This synergistic management of space and shear,

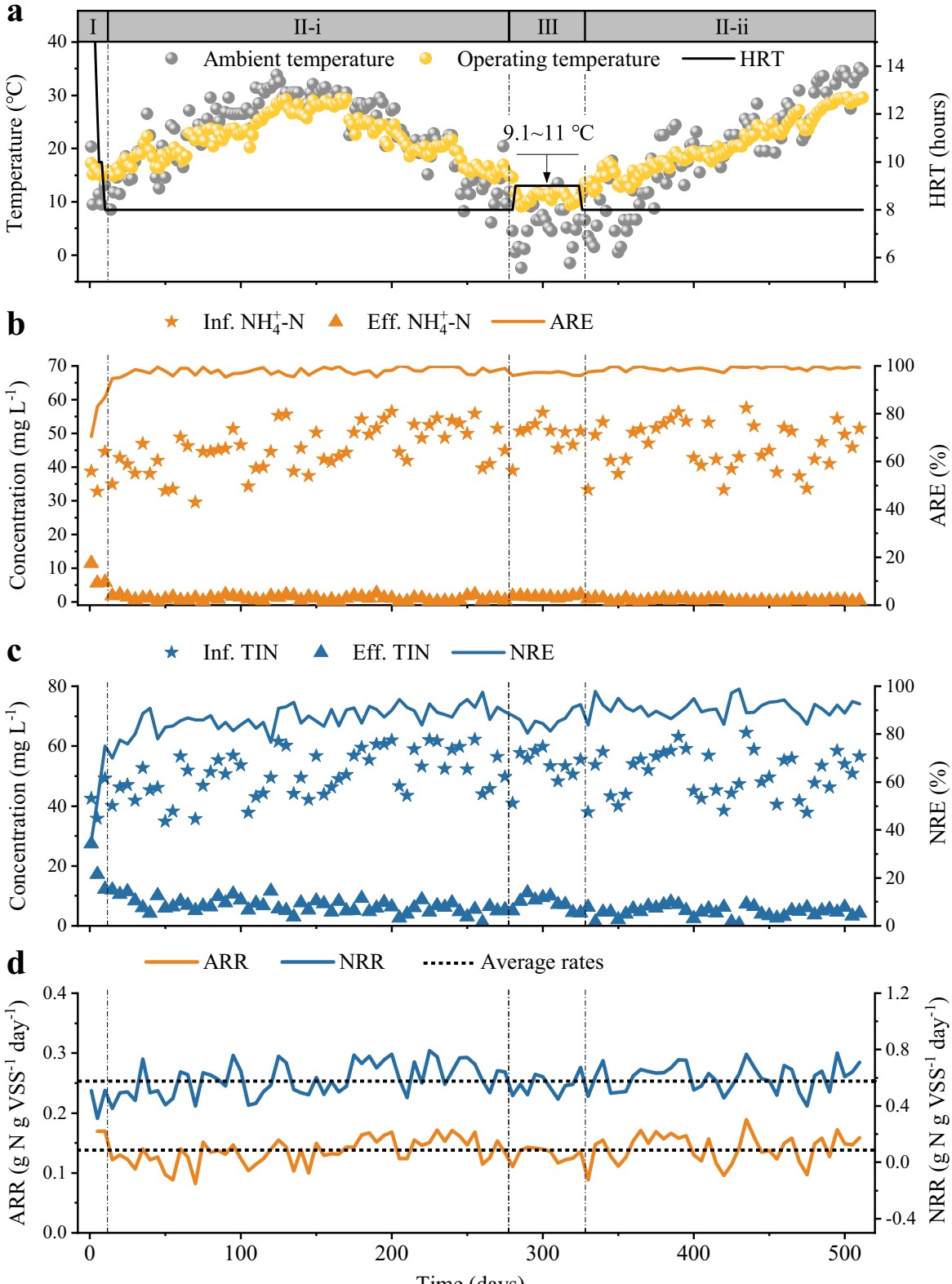

**Fig. 2 | Long-term nutrient removal performance of the A/O-MBBR system with V-carriers. a** Operational timeline showing hydraulic retention time (HRT) and daily operating temperature (recorded at 11:00 AM). Phases: I (start-up), II-i (suitable temperature), III (low temperature), II-ii (suitable temperature). **b** Influent (Inf.) and effluent (Eff.) ammonium nitrogen (NH₄⁺-N) concentrations with removal efficiency (ARE). **c** Influent (Inf.) and effluent (Eff.) total inorganic nitrogen (TIN) concentrations with removal efficiency (NRE). **d** Ammonia (ARR) and TIN (NRR) removal rates. Data points represent measurements taken every 4 days over 512 days of continuous operation.

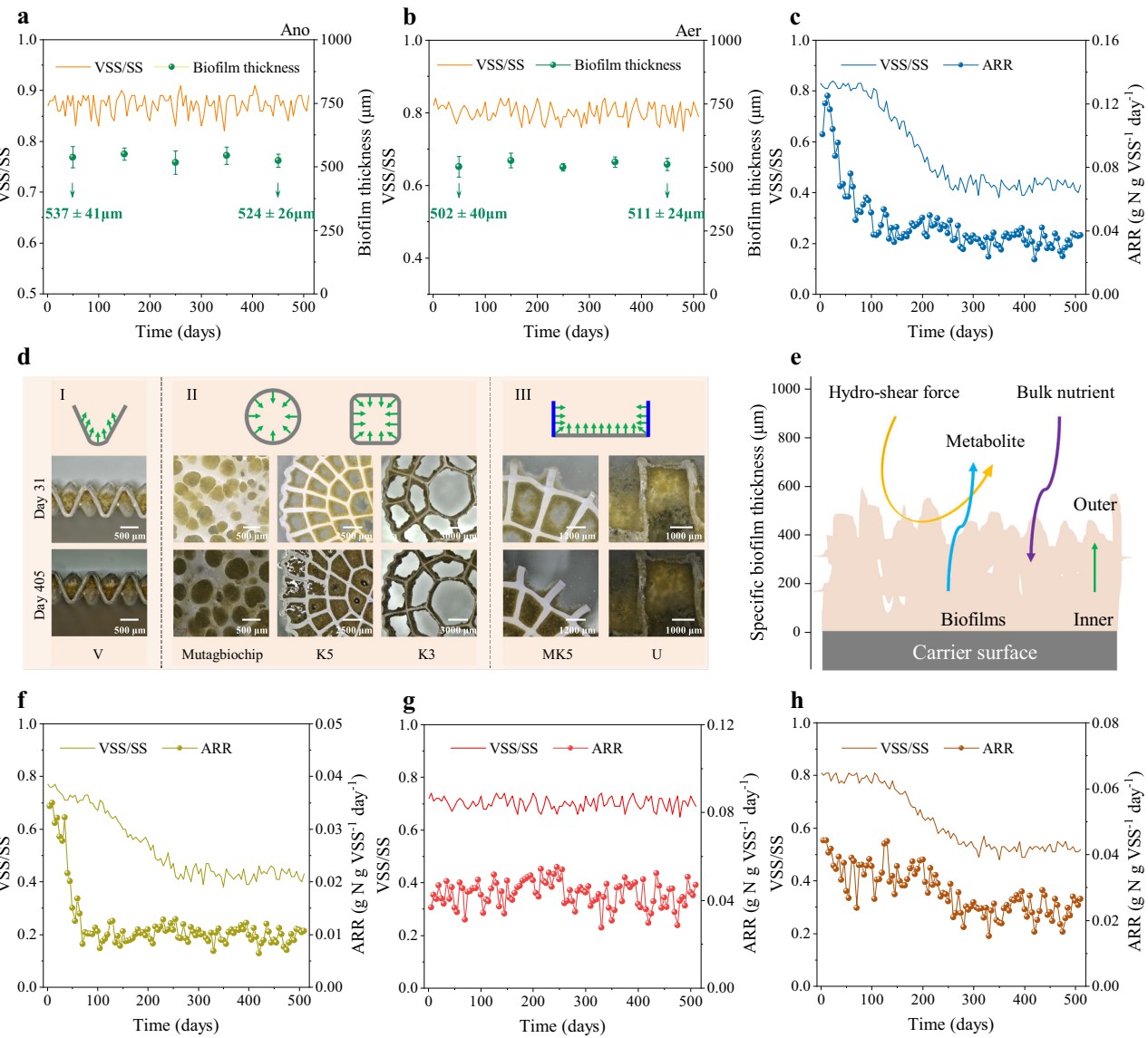

**Fig. 3 | Key factors for stable and efficient mass transfer in V-carriers.** Biofilm volatile suspended solids (VSS) to suspended solids (SS) ratio and thickness of V-carriers during long-term operation under anoxic (**a**) and aerobic (**b**) conditions. The VSS/SS ratio is a key indicator of biofilm health; a high, stable ratio signifies a biofilm with minimal inorganic scaling and high metabolic activity. Aerobic biofilm VSS/SS ratio and ammonia removal rate (ARR) for Mutagbiochip (**c**), K5 (**f**), K3 (**g**), and U (**h**) carriers. The declining VSS/SS ratio in panels c, f and h indicates inorganic accumulation and clogging. **d** Schematic diagram of biofilm growth patterns and corresponding light micrographs (days 31 and 405) for V, Mutagbiochip, K5, K3, MK5, and U carriers under aerobic conditions. **e** Proposed mechanism of hydraulic shear-driven biofilm self-cleaning. Data represent mean ± s.d. ($n = 3$ technical replicates). Schematics (**d**, **e**) were created using Microsoft PowerPoint 2021.

which we term the hydro-topological strategy, integrates precise topological guidance for hydraulic exposure with sustained shear for autonomous regulation to maintain an optimal, stable biofilm.

### Biofilm dynamics and functional adaptation to temperature

To elucidate the microbial basis for the observed operational stability, we investigated the physiological and spatial dynamics of V-carrier biofilms in response to temperature fluctuations (Fig. 4). Ex situ nitrification potential exhibited a strong positive correlation with temperature ($P = 7.12 \times 10^{-35}$, $R^2 = 0.591$), confirming its role as a primary regulator of microbial activity (Fig. 4a). Notably, the ex situ rates surpassed the in situ performance by factors of 1.55 (<15 °C) and 3.18 (>15 °C), indicating that the biofilms possessed intrinsic metabolic capacity that was not fully expressed under the reactor's substrate-limiting and dynamically fluctuating operational conditions (e.g., temperature, pH, dissolved oxygen). Quantitative PCR revealed rapid

enrichment of ammonia-oxidizing (AOB) and nitrite-oxidizing bacteria (NOB), with their combined absolute abundance stabilizing at $(1.0 \pm 0.2) \times 10^{12}$ copies per gram of biofilm VSS after 150 days of operation (Fig. 4b). This high and stable abundance of nitrifiers confirms the effectiveness of the V-carrier's microenvironment in supporting functional community retention. Spatial analysis provided further mechanistic insights. The relative abundance of nitrifiers in the community was also positively correlated with temperature ($P = 0.0091$, $R^2 = 0.542$). Fluorescence in situ hybridization (FISH) visualization aligned with this trend, showing stronger signals for AOB and NOB at higher temperatures (Fig. 4c, d). More importantly, a distinct spatial reorganization was observed: nitrifiers predominantly occupied the inner biofilm layers at higher temperatures but distributed more uniformly throughout the biofilm matrix during seasonal cooling. This redistribution is interpreted as an adaptive strategy to mitigate diffusion limitations and maintain metabolic activity under

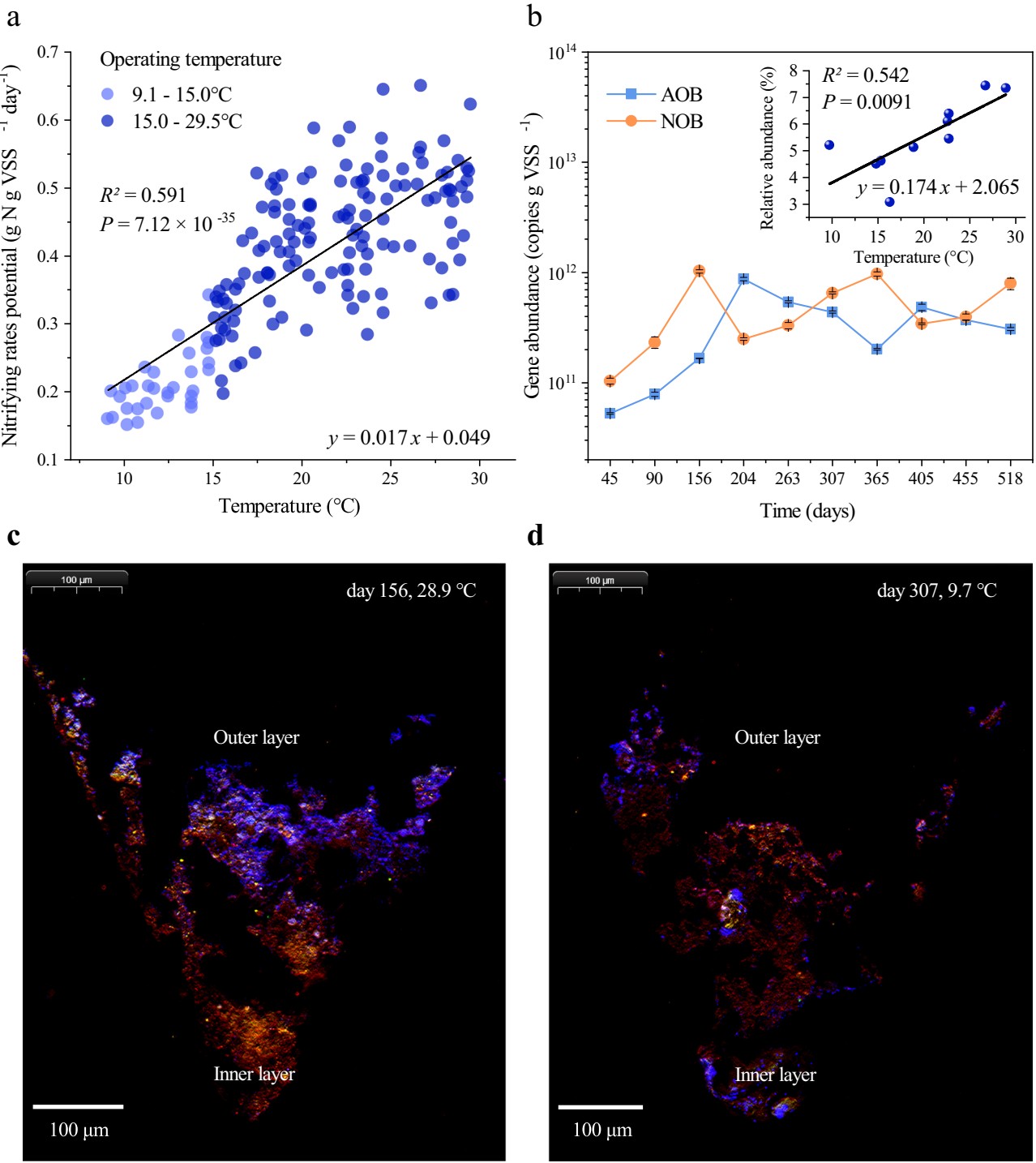

**Fig. 4 | Biofilm dynamics and spatial distribution in V-carriers during long-term operation. a** Ex situ nitrification rates of V-carriers across temperatures (9.1–29.5 °C). Linear regression (solid line) with 95% confidence interval ($P = 7.12 \times 10^{-35}$, two-tailed $t$-test). **b** Absolute abundances of ammonia-oxidizing (AOB) and nitrite-oxidizing (NOB) bacteria quantified by qPCR (left y-axis). Inset shows relative abundance correlation with temperature analyzed by two-tailed Pearson correlation ($R^2 = 0.542$, $P = 0.0091$). Data represent mean ± standard deviation ($n = 3$ independent biological replicates). **c, d** Fluorescence in situ hybridization combined with confocal laser scanning microscopy (FISH-CLSM) images. **c** Biofilm on day 156 (28.9 °C) showing AOB (green) and NOB (red) co-localization in inner layers. **d** Biofilm on day 307 (9.7 °C) showing uniform distribution of nitrifiers under low-temperature conditions. Images are representative of $n = 6$ independent experiments with similar results. Scale bars: 100 μm (**c, d**). Total bacteria were stained with DAPI (blue) and orange indicates co-localization of AOB and NOB signals (**a, b, c**, and **d** were assembled and annotated in OriginPro 2021).

low-temperature stress. Furthermore, the persistent presence of internal voids within the biofilms, irrespective of temperature (Supplementary Fig. 7), is consistent with active cell lysis and material export[37–39]. This is definitely supported by live-dead staining (Supplementary Fig. 8), which reveals a higher proportion of dead cells and lysing debris in the inner biofilm layers, illustrating a gradient from the interior to the bulk liquid. This visual evidence confirms that the voids are pathways formed by continuous cell lysis and subsequent export of cellular material. This morphological feature provides physical evidence of an ongoing self-renewal process, underpinning the dynamic

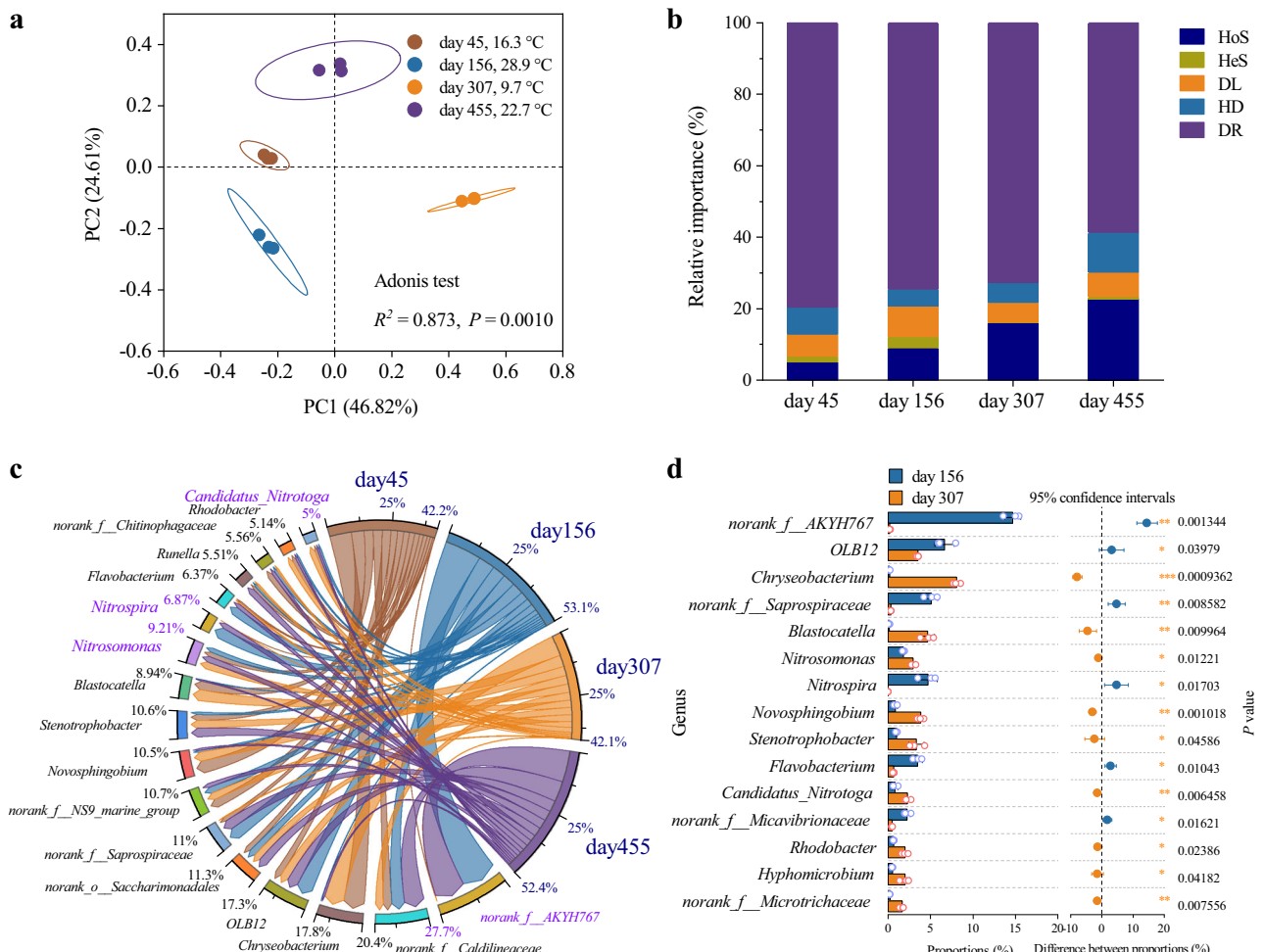

**Fig. 5 | Microbial community characteristics of V-carrier biofilms. a** Principal coordinates analysis (PCoA) of ASV-level community composition based on Bray–Curtis dissimilarity. The analysis was performed on n = 3 biologically independent samples. Ellipses represent 95% confidence intervals for each phase. Community differences were assessed using a two-tailed Adonis test ($R^2$ = 0.873, $P$ = 0.001). **b** iCAMP analysis quantifying the relative importance of ecological processes governing community assembly: homogeneous selection (HoS), heterogeneous selection (HeS), dispersal limitation (DL), homogeneous dispersal (HD), and drift (DR). **c** Circos plot showing genus-level community composition dynamics across operational phases. Ribbon connections highlight dominant taxa persistence. **d** Welch's two-tailed $t$-test bar plot for the core genera of V-carrier biofilms at 28.9 °C (day 156; n = 3 biologically independent samples) and 9.7 °C (day 307; n = 3 biologically independent samples). Error bars represent the standard deviation (s.d.). Differences in the proportion of core genera are presented as mean ± s.d. Exact $P$ values are indicated directly on the figure. Statistical significance is denoted as follows: * 0.01 < $P$ ≤ 0.05, ** 0.001 < $P$ ≤ 0.01, *** $P$ ≤ 0.001.

self-cleaning mechanism intrinsic to the V-carrier design. Collectively, these findings demonstrate that the V-carrier fosters a resilient biofilm ecosystem capable of physiological, spatial, and compositional adaptation to maintain function against environmental perturbations.

## Microbial community assembly and functional resilience

We analyzed the bacterial community structure and succession in aerobic V-carrier biofilms via 16S rRNA gene amplicon sequencing to link ecosystem-level processes to system function (Fig. 5). Principal coordinates analysis revealed that operating temperature was the primary driver of community dynamics (Adonis test, $R^2$ = 0.873, $P$ = 0.001) (Fig. 5a). Furthermore, the strong fit to the neutral community model ($R^2$ > 0.5 across 9.1–29.5 °C) indicated that stochastic processes (e.g., growth, apoptosis, drift) governed initial biofilm assembly and the self-cleaning process (Supplementary Fig. 9). An iCAMP analysis quantified the ecological processes, confirming that while stochasticity dominated assembly, temperature was the key deterministic factor modulating succession (Fig. 5b). Microbial diversity and richness increased throughout the operation (Supplementary Table 2). These results illustrate a dual regulatory mechanism: stochastic processes maintain a self-cleaning, stable community

foundation, while deterministic temperature shifts guide its functional succession.

At the genus level, the core nitrifying guild consistently comprised *Nitrosomonas* (AOB), *Nitrospira* (NOB), and *Candidatus* (*Ca.*) *Nitrotoga* (NOB). Their abundances shifted markedly with temperature (Fig. 5c). During cooling, *Nitrosomonas* increased from 1.85% to 2.92%, while *Nitrospira* decreased from 4.67% to below detection limits. Concurrently, the cold-tolerant NOB *Ca. Nitrotoga* increased significantly from 0.81% to 2.28% ($P$ = 0.006) (Fig. 5d), demonstrating a compensatory niche replacement that underpinned the functional resilience of nitrification observed during low-temperature operation (Fig. 2). The most pronounced shift occurred in the putative denitrifier *norank_f_AKYH767*[40,41], which declined dramatically from 14.8% to 0.2% ($P$ = 0.0013) with cooling. This decline likely resulted from combined pressures of limited substrate access and reduced enzymatic activity at low temperatures, potentially exacerbated by the outward migration of nitrifiers within the biofilm (Fig. 4d), a shift that may be associated with elevated dissolved oxygen concentrations in the inner biofilm layers to sustain nitrification, thereby creating a less favorable microenvironment for denitrifiers. This could further limit nitrate diffusion to inner-layer denitrifiers. Thus, the V-carrier system maintains

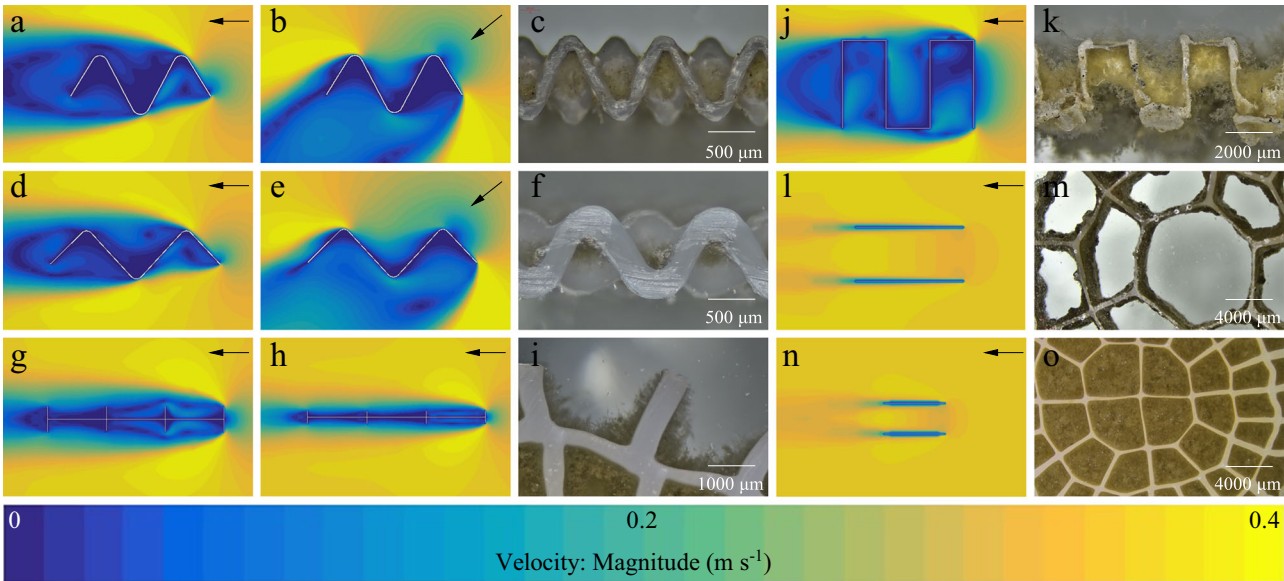

**Fig. 6 | Computational fluid dynamics (CFD)-simulated flow fields and biofilm morphology of carriers after 405 days of operation.** Flow fields around a V-carrier with an opening ratio of 1.39 (1.0/0.72) under **a** parallel and **b** 45° flow incidence. **c** Biofilm morphology on the stably-operated V-carrier with opening ratio 1.39. Flow fields around a V-carrier with an opening ratio of 2.08 (1.5/0.72) under **d** parallel and **e** 45° flow incidence. **f** Biofilm morphology on the stably-operated V-carrier with opening ratio 2.08. Flow fields (parallel flow) around carriers with cell dimensions of 2.3 × 2.3 mm and ridge heights of (**g**) 1.0 mm and (**h**) 0.5 mm. **i** Biofilm morphology on the MK5 carrier. **j** Flow field (parallel flow) and **k** biofilm morphology of the U-carrier. **l** Flow field (parallel flow) and **m** biofilm morphology of the K3 carrier. **n** Flow field (parallel flow) and **o** biofilm morphology of the K5 carrier.

functional stability through a multi-faceted strategy: it selects for beneficial, adaptive taxa (e.g., *Ca. Nitrotoga*), reduces competition from less robust organisms, and its hydro-topological design ensures efficient mass transfer even under metabolically challenging conditions.

## Discussion
### A hydro-topological design paradigm
The propensity of conventional MBBR carriers to clog represents a fundamental limitation that undermines their sustainability benefits. While the critical importance of biofilm thickness regulation has been recognized[20,42,43], prevailing design strategies have relied on geometric confinement—exemplified by the recently developed Z-carrier, which postulated that grid height alone could control biofilm thickness[17]. Our systematic comparison using U-shaped and MK5 carriers invalidates this premise. We demonstrated that ridge height, analogous to grid height, fails to dictate biofilm thickness: on the MK5 carrier, biofilms did not reach the ridge height due to excessive shear, while the U-carrier experienced clogging as biofilms overgrew the ridge surfaces (Fig. 3d-III). This outcome, confirmed by CFD simulation of the flow field, indicated a 56–91% higher hydraulic shear stress in the center of the grid cell compared to the area near the ridges, leading to significantly thinner biofilms in the central region and the failure to maintain a stable target biofilm thickness (Fig. 6g–k). This reliance on simplistic geometric assumptions has constrained the development of advanced biofilm systems. Our work necessitates a fundamental re-evaluation of carrier design principles, shifting the paradigm from passive geometric confinement to active hydro-topological control that integrates hydraulic conditions with biofilm ecology.

This study establishes that effective biofilm regulation requires the synergistic management of hydraulic shear and biofilm metabolic dynamics. We identify three pillars for designing sustainable MBBR carriers. First, constraining biofilm thickness is imperative for mass transfer efficiency. Even in the initially unclogged K5 carrier, the nitrification rate decreased by 61% as the biofilm thickened from $368 \pm 29\,\mu m$ to $791 \pm 44\,\mu m$. This decline occurred within a critical 35-

day period (days 20–55), directly linking biofilm thickness to functional performance (Fig. 3f). Second, thickness control must be integrated with an autonomous self-cleaning capability. The failure of the Mutagbiochip carrier underscores that equating carrier thickness with controllable biofilm thickness is a critical flaw; it ignores the intrinsic metabolic activity of biofilms (Fig. 3c). Biofilms are dynamic ecosystems characterized by cell lysis, metabolite export, and material recycling[37,38]. Designs that neglect these processes create an imbalance between growth and decay, inevitably leading to clogging, as seen in the Mutagbiochip, K5, and U carriers (Fig. 3). Third, and most critically, sustained hydraulic shear on the biofilm surface is the linchpin for effective self-cleaning. In tubular carriers, metabolite clearance is governed by internal flow dynamics. CFD simulations under identical hydraulic conditions revealed that a smaller tube diameter results in lower internal flow velocity, which in turn leads to reduced shear stress (Fig. 6l-o), a phenomenon directly supported by the Hagen-Poiseuille equation[26–28]. This explains why carriers with small hollows (e.g., Mutagbiochip, K5) clog, while those with larger hollows (e.g., K3) maintain patency—the biofilm growth has a negligible impact on the internal hydraulics of larger tubes (Fig. 3d). The V-carrier circumvents this limitation entirely by its open topology, which ensures direct hydraulic exposure of the biofilm surface, thereby enabling real-time metabolite clearance and efficient mass transfer.

Based on these principles, we propose a hydro-topological design framework for next-generation MBBR carriers. The V-carrier is a manifestation of this framework, where the opening ratio governs shear intensity and the groove depth sets the maximum biofilm thickness. Validation tests using a V-carrier with a larger opening ratio (1.5/0.72 = 2.08) confirmed a significantly thinner biofilm (~240 μm) (Fig. 6f). Crucially, CFD simulations revealed a ~44% downward shift of the iso-velocity contours within the groove, which correlated with the observed biofilm reduction, demonstrating that hydro-topological parameters (hydraulic shear, opening ratio and groove depth) exert absolute control over biofilm thickness by dictating the hydraulic shear environment (Fig. 6a-f). The adjustability of these parameters allows for tailoring carriers to specific wastewater characteristics and

treatment objectives. More broadly, this study provides a generalizable methodology for carrier optimization. By employing targeted in situ and ex situ analyses, the biofilm composition and metabolic state can be characterized for any design, enabling the data-driven development of customized MBBR solutions. This framework moves the field beyond empirical trial-and-error towards a principled engineering science of biofilm carriers, with profound implications for the sustainability of wastewater treatment infrastructure.

## Advantages and application outlook

This study demonstrates a pure biofilm system employing the hydro-topologically optimized V-carrier that achieves efficient, stable nutrient removal under low-temperature and high-loading conditions, showcasing its potential for energy-efficient mainstream wastewater treatment. The system attained high specific removal rates of $0.138 \pm 0.022 \, \text{g N g VSS}^{-1} \, \text{day}^{-1}$ (ammonia) and $0.587 \pm 0.108 \, \text{g N g VSS}^{-1} \, \text{day}^{-1}$ (TIN), signifying exceptional biomass-specific activity. More significantly from an engineering perspective, it achieved exceptional volumetric nitrifying rate potentials of $335.61 \, \text{g N m}^{-3} \, \text{day}^{-1}$ (<15 °C) and $688.54 \, \text{g N m}^{-3} \, \text{day}^{-1}$ (>15 °C), performance competitive with established advanced technologies[44–47]. Remarkably, this high performance was achieved at a carrier filling ratio of just 30% in the aerobic reactor, suggesting potential for further capacity enhancement. By also achieving complete biomass retention and eliminating the need for sludge recirculation, this system presents a compelling solution for wastewater treatment plants confronting stringent discharge standards, spatial limitations, and pressing energy-saving mandates.

A key finding is that treatment efficiency is decoupled from sheer biomass quantity. During stable operation, the biofilm VSS concentration in the V-carrier system ($1569 \pm 96 \, \text{mg L}^{-1}$) was 44% lower than that in the K3 carrier system ($2807 \pm 165 \, \text{mg L}^{-1}$) (Supplementary Fig. 10). Remarkably, this lower biomass supported a 3.2-fold higher nitrification rate in the V-carrier system compared to the K3 system (Fig. 2d). This underscores that efficient microbial cooperation within a low-biomass, structurally optimized biofilm ecosystem can yield superior nutrient removal, highlighting the ecological advantage conferred by our design. The hydro-topological strategy established here opens avenues for optimizing other biofilm-based processes, such as anaerobic ammonium oxidation (anammox), partial denitrification, and simultaneous nitrification-denitrification[48]. By fundamentally reducing mass transfer resistance and mitigating detrimental microbial competition—common challenges in conventional systems using suspended sludge, granular biofilms, or non-optimized carriers[49–51]—our approach enables precise management of microbial communities. This provides a paradigm for enhancing these promising technologies.

Beyond its immediate application, this work establishes an ideal experimental platform for probing fundamental questions in biofilm science and microbial ecology. A pressing challenge is elucidating nitrous oxide ($N_2O$) emission mechanisms from biofilm processes, given their significant impact on the carbon footprint of wastewater treatment[52–54]. The stable, well-defined niche created by the V-carrier, combined with its low-biomass optimization, offers an unprecedented opportunity to disentangle the complex metabolic pathways governing $N_2O$ production and consumption. Furthermore, this system is uniquely suited for investigating the dynamics and functions of extracellular polymeric substances (EPS). While EPS components are identified, their production kinetics, functional roles, and interplay with metabolic activity remain poorly quantified, hindering their integration into predictive biofilm models[55,56]. The V-carrier platform provides the controlled microenvironment necessary to address these fundamental gaps, paving the way for more accurate modeling and a deeper understanding of biofilm ecology.

From an engineering perspective, the hydro-topological design of the V-carrier translates into significant practical advantages for large-scale implementation. Pilot-scale fluidization tests demonstrated the V-carrier's superior hydrodynamic efficiency: it requires over 30% less aeration intensity than conventional cylindrical carriers for complete mixing, a result of the asymmetric flow around its grooves generating high drag for efficient motion, unlike the balanced pressure and minimal torque from the cross-flow in cylindrical carriers which leads to less efficient movement (Supplementary Fig. 11). This confirms that the V-carrier's design directly addresses a major operational cost factor. Furthermore, long-term pilot operation treating real municipal wastewater over 400 days confirmed the system's robust stability, with no carrier clogging or performance decay observed (Supplementary Fig. 12). The V-carriers maintained excellent physical integrity, with only slight wear at the corners of the anoxic carriers attributed to mechanical stirring—an issue being addressed by redesigning the blades. This successful validation underscores the V-carrier's potential for energy-efficient and reliable large-scale application.

# Methods

## Reactor configuration and operation

A laboratory-scale anoxic/aerobic (A/O) MBBR system was constructed using customized polypropylene tanks. The working volumes were 150 L (anoxic zone), 250 L (aerobic zone), and 20 L (secondary clarifier). The system was operated in a pure biofilm mode (no sludge recirculation) for over 500 days, treating prepared municipal wastewater. Nitrified liquor was recirculated from the clarifier to the anoxic zone at a rate of 300% relative to the influent flow. The volumetric filling ratios of carriers were maintained at 16% and 30% in the anoxic and aerobic reactors, respectively. The V-carriers were provided by Hangzhou Tao of Water Technology Co., Ltd (Hangzhou, China). The methodology for determining the carrier's number, bulk density and structural porosity is provided in Supplementary Equations. The system schematic can be found in Supplementary Fig. 1.

## Wastewater characteristics and feed preparation

Raw municipal wastewater was collected daily from the septic tank effluent of Shangkun Park (Hangzhou, China). Tap water, aerated for 24 h to remove residual chlorine, was used to dilute this primary effluent to simulate typical influent concentrations for a local wastewater treatment plant (Chengxi WWTP, Hangzhou, China), resulting in a feed containing $143.8 \pm 23.4 \, \text{mg L}^{-1}$ soluble chemical oxygen demand (sCOD), $45.9 \pm 6.7 \, \text{mg L}^{-1} \, NH_4^+$-N, and $50.7 \pm 7.4 \, \text{mg L}^{-1}$ total inorganic nitrogen (TIN), with an initial C/N ratio of $2.8 \pm 0.3$. Phosphorus was present predominantly as orthophosphate (>99% of total phosphorus) at a concentration of $4.8 \pm 1.7 \, \text{mg L}^{-1}$. The hydraulic retention time (HRT) was managed to comply with the discharge standards for major water pollutants mandated by Zhejiang Province, China (see Supplementary Table 3 for detailed standards and wastewater characteristics). To enhance denitrification and meet the designed target, sodium acetate was supplemented into the anoxic zone, with the resultant C/N ratio being maintained at $4.9 \pm 0.3$ (Supplementary Fig. 2).

## Biofilm establishment and reactor startup

Prior to long-term operation, clean V-carriers were added to the reactors and fluidized to ensure uniform distribution. Biofilm cultivation was initiated by feeding the reactors with activated sludge (mixed liquor suspended solids concentration = $3512 \, \text{mg L}^{-1}$) obtained from the Chengxi Wastewater Treatment Plant (Hangzhou, China). During the first three days, 50 L of activated sludge was introduced twice daily at an HRT of 24 h. The HRT was then progressively reduced by increasing the inflow rate until stable nutrient removal performance was achieved, marking the start of the long-term experimental phase. The detailed Biofilm Establishment and System Startup Protocol is provided in Supplementary Note 2.

## Analytical methods

**Water quality monitoring.** Influent and effluent samples were collected daily, filtered through 0.45-μm membrane filters (NAVIGATOR, China), and immediately analyzed. Concentrations of sCOD, $NH_4^+$-N, TIN, and $PO_4^{3-}$-P were determined spectrophotometrically (GNST-900S, China) using commercial reagent kits (Catalog Number G0402, Genesite®, China) according to the manufacturer's protocols. Temperature, dissolved oxygen (DO), and pH were monitored using multiparameter probes (HC2200, USA).

**Biofilm characterization.** Biofilm samples were harvested from carriers by gentle brushing with a sterile brush, followed by centrifugation. The resulting pellets were stored at −80 °C for subsequent microbial analysis. Suspended solids (SS) and volatile suspended solids (VSS) were measured in accordance with APHA Standard Methods[57]. The elemental composition (e.g., for scaling analysis) was determined using inductively coupled plasma mass spectrometry (ICP-MS, iCAP RQ, Thermo Fisher Scientific). For morphological analysis, freshly harvested carriers were immediately suspended in deionized water and examined using an industrial light microscope (GP-304K, China) to characterize structure and measure thickness. Specific removal rates for ammonia (ARR), TIN (NRR), and sCOD (CRR) were calculated on a per-gram of biofilm VSS basis. Detailed procedures for SS, VSS measurement, elemental analysis, and rate calculations are provided in Supplementary Note 3 and Supplementary Equations.

## Microbiological analysis

**Nucleic acid extraction and quantification.** Total genomic DNA was extracted from frozen biofilm biomass samples using the E.Z.N.A.® Soil DNA Kit (Catalog Number D5625-01, Omega Bio-tek, USA) according to the manufacturer's instructions, under aseptic conditions. The concentration and quality of the extracted DNA were assessed using a NanoDrop 2000 spectrophotometer (Thermo Scientific, USA) and verified by 1.0% agarose gel electrophoresis. Qualified DNA samples were stored at −80 °C until further analysis.

**Quantitative PCR (qPCR).** The absolute abundances of the 16S rRNA gene (total bacteria), and functional genes specific to ammonia-oxidizing bacteria (AOB, *amoA*) and nitrite-oxidizing bacteria (NOB) were quantified by real-time qPCR. All reactions were performed in triplicate. Primer sequences and detailed thermal cycling conditions are provided in Supplementary Table 4.

**16S rRNA gene amplicon sequencing.** Bacterial community analysis was performed via high-throughput sequencing of the 16S rRNA gene. For each selected time point, genomic DNA from three independent biofilm samples ($n = 3$) was used as template. The V3–V4 hypervariable regions were amplified using primers 341F and 806R[58] on a T100 Thermal Cycler (Bio-Rad, USA). Amplification and library preparation followed established protocols. Sequencing was conducted on an Illumina MiSeq platform (Majorbio Bio-Pharm Technology Co., Ltd., Shanghai, China). Raw sequence data have been deposited in the China National Center for Bioinformation (CNCB) database under BioProject accession number CRA032524. Detailed bioinformatics processing, including quality filtering, ASV clustering, and taxonomic assignment, is described in Supplementary Note 4.

**Fluorescence in situ hybridization (FISH).** The spatial distributions of AOB and NOB within biofilms were analyzed using FISH combined with confocal laser scanning microscopy (FISH-CLSM). Biofilm samples were collected from aerobic V-carriers on day 156 (28.9 °C) and day 307 (9.7 °C). Hybridizations were performed using Cy3-labeled oligonucleotide probes NSO190 (for most AOB) and Ntspa712E (for the genus *Nitrospira*)[32], with DAPI used for total cell staining. Detailed FISH

protocols, including fixation, hybridization conditions, and microscopy parameters, are provided in Supplementary Note 5.

**Live-dead cell staining of biofilms.** The viability and spatial distribution of cells within the biofilms were assessed using a dual fluorescent staining method based on SYTO 9 and propidium iodide (PI), followed by CLSM imaging. Carrier samples with attached biofilms were collected on day 409 (14.1 °C) and analyzed. The comprehensive staining and imaging protocol is provided in Supplementary Note 6.

## Ex situ assessment of nitrifying potential

The temperature-dependent nitrifying activity of biofilms grown on V-carriers was assessed periodically via ex situ batch assays conducted every two days throughout the operational period. Assays were performed in a 3-L transparent polypropylene batch reactor equipped with a bottom air diffuser. The reactor was pre-calibrated with tap water to ensure complete fluidization of the carriers under aeration. The reaction medium was a synthetic wastewater (composition detailed in Supplementary Table 5) devoid of organic carbon to selectively assess nitrification. For each assay, 300 V-carriers were aseptically retrieved from the aerobic zone of the main A/O-MBBR system and transferred to the batch reactor containing 3 L of the synthetic medium. The assay conditions were rigorously controlled: temperature was maintained at the recorded in-situ value using a water bath and monitored continuously with a multiparameter sensor (HC2200, USA); dissolved oxygen (DO) was kept at $4.5 \pm 0.5$ mg $L^{-1}$; and pH was maintained at 7.5 through the continuous addition of a sodium bicarbonate solution ($NaHCO_3$, 150 g $L^{-1}$) at a rate of 0.1 mL $min^{-1}$ to stabilize alkalinity. Liquid samples were collected at 0, 30, 60, 120, and 180 min for immediate analysis of $NH_4^+$-N and $NO_2^-$-N concentrations. The final $NH_4^+$-N concentration was set above 15 mg $L^{-1}$ to preclude substrate limitation during the assay. The specific nitrification potential (expressed as g N g $VSS^{-1}$ $d^{-1}$) was calculated from the linear rate of ammonium depletion, as detailed in Supplementary Eq. (8). Following each assay, the carriers were promptly returned to the aerobic reactor of the main system. Control assays (without carriers) were conducted in parallel to account for any abiotic reactions.

## Computational fluid dynamics (CFD) analysis

To provide a quantitative, mechanistic understanding of the hydraulic environments, we conducted comprehensive CFD simulations of the flow fields and shear stresses on the carrier surfaces. The simulations were performed using the finite volume method (FVM) implemented in the commercial software Siemens STAR-CCM+ (version 2406.19.04.007). The fluid flow was modeled as a three-dimensional, incompressible system governed by the continuity equation and the Reynolds-averaged Navier–Stokes (RANS) equations, with turbulence closure achieved using the *Realisable k-ε* model. All computations were executed on a high-performance workstation (Intel Xeon Gold 6242 R CPU, 3.10 GHz, 20 cores) operating in parallel mode. The detailed geometrical models, boundary conditions, and validation of the CFD setup are provided in Supplementary Fig. 13 and Supplementary Note 7.

## Statistics and reproducibility

Statistical analyses were performed using OriginLab 2021 (OriginPro, OriginLab Corporation, Northampton, MA, USA) and Microsoft Excel 2021 (Microsoft Corporation, Redmond, WA, USA). Microbiological data were analyzed on the Majorbio Cloud Platform (https://cloud.majorbio.com). Experimental data are presented as mean ± standard deviation (s.d.). Specific statistical tests applied included the two-tailed paired *t*-test, Welch's two-tailed *t*-test, two-tailed Pearson correlation, and the two-tailed Adonis test. A *P* value of less than 0.05 was considered statistically significant, with the significance level ($\alpha$) set at

0.05 for all tests. Data points below the detection limit were assigned a value of zero for analysis purposes. No data were excluded from the analyses unless otherwise specified. No statistical method was used to predetermine sample size. The experiments were not randomized. The investigators were not blinded to allocation during experiments and outcome assessment.

## Reporting summary

Further information on research design is available in the Nature Portfolio Reporting Summary linked to this article.

## Data availability

The data generated in this study are provided within the article and its Supplementary Information. Raw sequence data have been deposited in the China National Center for Bioinformation (CNCB) database under BioProject accession number CRA032524 and are publicly accessible at: https://ngdc.cncb.ac.cn/gsa/browse/CRA032524. Source data are provided with this paper.

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

## Acknowledgements

This research was supported by National key R&D Program of China (2024YFD2400100; Y.L.), the scientific research project in Fengjiawan of Hainan Provincial Seed Industry Laboratory (B24H10035; Y.L.), the National Natural Science Foundation of China (Grant No. 31971794; K.S.), the Fundamental Research Funds for the Central Universities 226-2025-00074; Y.L., China Agriculture Research System of MOF and MARA (CARS-49; Y.L.), Key Research & Development Projects of Zhejiang Province (2025C02203; K.S.), and Student Research and Entrepreneurship Project of Zhejiang University (2023; K.S.).

## Author contributions

Y.F., Y.L., and K.S. conceived the idea and designed the research. Y.F. and Z.Z. conducted the experiments, collected and interpreted the data. Y.F., Z.Z., B.X., and K.S. prepared the figures, wrote and edited the manuscript and supplementary information. Y.L. and K.S. jointly oversaw the administration of the project and supervised the work. All the authors discussed the results and commented on the paper.

## Competing interests

The authors declare no competing interests.
