## [Transparent Peer Review file · Nature Communications]

A hydro-topological strategy enables self-regulating biofilms for sustainable wastewater treatment

Corresponding Author: Professor kuichuan sheng

Version 0:

Reviewer comments:

Reviewer #1

(Remarks to the Author)

The manuscript's main focal area entails biofilm technology development for water resource recovery, including two key notions; the meaning of which needed to be defined by co-authors in the ensuing text, i.e.,

Hydro-topological strategy: they define it as a reactor "design strategy", which seems to indicate the V-shape of the developed and tested carrier media.

Self-regulating biofilm: they describe it as the hydraulic behaviour of the new carrier design is intended to serve the purpose of more efficient biofilm sloughing from the media.

- What are the noteworthy results? ANSWER: A new type of biofilm carrier was tested, however, the advantages of this carrier – over state-of-the-art technology, e.g., zeta-carrier – were not sufficiently supported by new evidence. In fact, there is not even a photograph of the carrier shown only seemingly computer-generated images.
- Will the work be of significance to the field and related fields? ANSWER: As supported my below detailed evaluation and assess, my answer is NO, it will not.
- How does it compare to the established literature? If the work is not original, please provide relevant references. ANSWER: the manuscript establishes knowledge gaps based on the premises of erroneous understanding/interpretation of literature.
- Does the work support the conclusions and claims, or is additional evidence needed? ANSWER: I could not find any separate Conclusions as such. Meanwhile, some of the statements based on results do not support the e.g., high ammonia removal rate – benchmarked against literature data.
- Are there any flaws in the data analysis, interpretation and conclusions? - Do these prohibit publication or require revision? ANSWER: I am unsure about the data analysis. To me it is mostly down to the convenient way of interpreting literature in a way suitable for their objectives.
- Is the methodology sound? Does the work meet the expected standards in your field? ANSWER: It is partly sound. The main issue is the lack of benchmarking against the state-of-the-art technology and the lack of clear demonstration of novelty and benefits of the new technology.
- Is there enough detail provided in the methods for the work to be reproduced? ANSWER: clearly not –benchmark reference data is missing.

More detailed critical comments:

"inevitably clog. This clogging severely diminishes treatment efficiency" Very ambiguous and seemingly incorrect statement – right in the problem statement of the abstract. Do you mean biofilm sloughing and removal of sloughed off biofilm?

"uncontrolled 29 biofilm accumulation. This results in severe carrier clogging and scaling, drastically diminishing the 30 effective surface area, impairing mass transfer and disrupting microbial community structure" Unclear and seemingly erroneous set of statements. Excessive growth on e.g., K1 carrier does NOT reduce the surface area. In fact the higher the biofilm thickness the lower the biofilm density, which effectively increases the effective diffusivity, i.e., f^*D_{water} . Also, nitrification rate is not expected to change much, only if normalised to the total gVSS on the carrier. Example: at 50 μm : 2 gN/(d.gTSS); at 200 μm : 0.6 gN/(d.gTSS).

There is no such thing as "disrupting community structure". Biofilm thickness has an impact on the microbial community structure, and higher thickness can select in microorganisms that e.g., harbour niche organisms with high micropollutant

biotransformation rate. So, it actually has advantages.

“clogged 32 carriers exhibit increased buoyant density, demanding higher energy inputs for mixing and 33 suspension”
Nope, it is right the opposite: biofilm density decreases with increased thickness, and not the other way around. In packed-bed filters, back-washing is used to remove/slough off excess biofilm biomass, which is accumulating in the filter – not in MBBRs. MBBRs had a filter media filling of around 40%, and no such behaviour can be observed.

“no existing MBBR carrier can reliably control biofilm thickness under the dynamic 40 hydraulic conditions” Erroneous identification of knowledge gap in literature: Kruger-VEOLIA’s zeta-carrier even comes with different mesh sizes that can control biofilm thickness between 50-500 μm .

“AnoxKaldnes introduced a carrier (the 'Z-carrier') based on a 42 geometric premise that its grid height would regulate biofilm thickness¹³.” And it does it very effectively, as opposed to what authors claim here. Importantly, Reference 13 has reported the innovation of the zeta-carrier using long-term experimentation, in which no such shortcomings were observed. In fact, authors of reference 13 also reported (Torresi et al., 2017, <http://dx.doi.org/10.1016/j.watres.2017.06.027>) on the systematic behaviour of the zeta-carrier in terms of in-biofilm diffusivity and sorption-desorption, which would not have been possible should the zeta-carrier allowed for irregular growth of biofilm thickness.

“However, their own 43 long-term evaluation revealed a critical flaw: biofilms at the center of the grid cells became 44 significantly thinner or were entirely absent²¹. This outcome underscores a fundamental stalemate⁴⁵ in sustainable carrier design, revealing a poor understanding of the hydrodynamic mechanisms that 46 control biofilm development. The failure of this simplistic geometric approach highlights the 47 necessity for a design paradigm rooted in hydraulic principles.” This seems to be a seriously flawed series of statements intended to undermine confidence in the state-of-the-art for the benefit of opening a knowledge gap on false premises.

1. It greatly misinterprets what has been reported in literature. Ref. 21 states “...had a clearly defined patch in the middle of each cell where the biofilm was significantly thinner, as a result of the carriers scraping each other.... Since the grid compartments were full of biomass up to the pre-determined wall heights, and calculations suggest that the maximum possible scraping depth from one Z-carrier into another Z-carrier compartment is 90 μm , the biofilm thickness on the four carrier types was determined to vary between 410–500, 310–400, 210–300 and 110–200 μm , respectively.”
2. Carriers scraping each other is not a sign of poor understanding of hydraulic behaviour.
3. It does not indicate that these carriers would “clog” in the way authors seem to mean it.

“The failure of this simplistic geometric approach ... The propensity for clogging in conventional tubular carriers” Co-authors’s may have limited understanding of the system they criticise. The zeta-carrier is NOT a tubular carrier but a three-dimensional wave-shaped flat surface covered with a mesh with a defined thickness.

“system exhibited high nutrient removal rates 120 of $0.138 \pm 0.022 \text{ g N g VSS}^{-1} \text{ day}^{-1}$ for ammonia” This observation is not benchmarked against literature evidence, and does not seem to be correct. For instance, reference 13 observed much higher removal rates using the zeta carrier of 50 μm , i.e., around $0.7 \text{ gN}/(\text{d.gVSS})$, and only thicker biofilms, i.e., above 200 μm showed similar but mostly higher values than that reported here.

“131 biofilm regulation—Hydro-topological synergy” “synergy” - a possible typo. Also, authors fail to thoroughly explain the meaning of the term they introduced.

Reviewer #2

(Remarks to the Author)

In this manuscript, the authors reported a new geometrical design for biofilm carriers used in moving bed biofilm reactor (MBBR). In contrast to the closed circular design, the authors used an open V-groove design, which was shown to lead to superior performance compared to the conventional design using various standard MBBR characterization. Microscopic analysis showed an open morphology of biofilm with holes, which the author attributed to cell lysis and contribute to biofilm self-renewal. Finally, metagenomics analysis was presented regarding the species distribution in such carrier microbial communities.

Overall, the research is performed with high quality, and the analysis is very solid. The presented technology has a strong potential to improve the existing technology. I support the publication of the manuscript in Nat. Comm. My only major concern is the lack of evidence of cell lysis, which is critical in the authors’ picture. Can the authors perform live-dead staining to definitely prove the existence of such process in the interior of the biofilm? Also, the author can do a better job in explaining some of the jargons for a broader audience. For example, in legend for Figure 3, what does VSS/SS ratio mean for the performance of the device?

Reviewer #3

(Remarks to the Author)

The manuscript presents a novel V-carrier design and provides a thorough scientific evaluation of its performance in wastewater treatment. The study offers valuable insights into biofilm self-regulation and demonstrates the potential advantages of the hydro-topological strategy. However, the manuscript would benefit from significant reorganization. The

overall structure and flow of ideas need improvement to enhance clarity and readability, as some sections currently lack logical sequencing despite the compelling presentation of results. Given these issues, I recommend major modifications before the manuscript can be considered for publication. Addressing the organization and ensuring a clearer narrative will greatly strengthen the impact of the work.

HERE ARE THE COMMENTS:

Introduction section

1. While the manuscript states that the MBBR process aligns with SDGs 6, 11, and 12, this section does not clearly discuss how MBBR operational characteristics contribute to these sustainability principles. The authors are encouraged to elaborate on the specific mechanisms through which MBBR supports these SDGs—for example, by detailing its impacts on resource efficiency, energy use, sludge reduction, land footprint, or resilience of sanitation systems. Providing this clarification would strengthen the justification of MBBR as a sustainable technology and improve the overall coherence of the sustainability narrative.
2. While the authors correctly identify clogging and uncontrolled biofilm accumulation as major limitations of conventional MBBR carriers, the manuscript does not provide any discussion of previous carrier designs reported in the literature or how those studies attempted to mitigate clogging. A concise yet critical overview of earlier design approaches—such as modified geometries, increased opening ratios, structured internal channels, or surface texturing—and an evaluation of their effectiveness in reducing fouling or improving hydrodynamics is essential. Furthermore, the manuscript should clarify the residual shortcomings of these designs that directly motivated the development of the new carrier proposed in this study. Addressing these points will strengthen the rationale for the new design and better situate the contribution of the current work within the context of prior research.
3. The manuscript could better contextualize the rationale for the current study by discussing previous carrier design strategies. Specifically, prior studies—including those employing CFD simulations—have proposed geometric or structural modifications to improve biofilm distribution, mass transfer, and reduce clogging. While such studies provide valuable indications and guidance for experimental design, they often lack long-term operational evaluation. Highlighting both the insights and the limitations of these previous approaches would strengthen the justification for developing and testing the new V-carrier in a long-term experimental setting.
4. The manuscript discusses the long-standing issue of clogging in MBBR systems but provides only a single example (the Z-carrier) as evidence of previous design limitations. This narrow focus unintentionally suggests that, over four decades of carrier development, no other design strategies or mitigation efforts have been explored. The authors are encouraged to expand this section by briefly summarizing additional carrier design attempts reported in previous studies and discussing their partial successes or remaining limitations. This broader context is essential to accurately position the Z-carrier as one example rather than the sole historical effort, and to more convincingly justify the need for the new design proposed in this work.

Methods Section:

1. Regarding the choice of using a square-shaped media instead of a cylindrical one, it would be useful to discuss the practical implications of this design choice. While the square shape may offer advantages in terms of maximizing surface area and potentially improving biofilm attachment, it is important to evaluate whether this design is as effective in real-world applications. Specifically, how does the square shape perform in terms of hydrodynamic behavior, ease of movement within the reactor, and resistance to clogging compared to cylindrical carriers? Further clarification and comparative analysis with cylindrical media would help assess its practical viability in large-scale wastewater treatment systems.
2. The authors diluted raw municipal wastewater with tap water but did not specify the dilution ratio. Additionally, residual chlorine and other ions in tap water may influence microbial activity, particularly nitrifiers and denitrifiers, potentially affecting the generalizability of the results to real WWTP conditions.
3. The manuscript reports adding 50 L of activated sludge twice daily during biofilm establishment. While this approach can accelerate biofilm colonization, the authors should clarify the rationale for the chosen frequency and volume, and discuss whether such repeated dosing could influence nutrient concentrations or microbial community development compared to typical operational conditions.
4. The manuscript describes the biofilm establishment protocol but does not clearly explain the rationale behind it. The authors should specify the principle guiding the chosen procedure—why activated sludge was added twice daily, how this promotes initial biofilm attachment, and the intended objective of this approach (e.g., accelerating colonization, ensuring uniform coverage, or stabilizing microbial communities). Providing this explanation would clarify the purpose of the protocol and strengthen the methodological section.

Results section

1. The manuscript reports a porosity value of 92% for the V-carrier; however, this statement is scientifically ambiguous and requires clarification. Because the carrier is fabricated from HDPE—a fully non-porous material—the reported porosity cannot correspond to material porosity. If the authors intend to describe structural or geometric porosity (i.e., the void volume fraction within the carrier's overall envelope), this must be explicitly stated. Moreover, the manuscript should clearly present the calculation method, including the definition of the total reference volume, how the void volume was determined (e.g., CAD-based volume analysis), and the exact formula used. Without a proper definition and methodological explanation, the reported porosity value is misleading and cannot be accurately interpreted in relation to the carrier's hydrodynamics or mass-transfer performance. The authors are encouraged to revise this section accordingly.
2. The manuscript claims that an opening ratio (length/depth) of 1.39 “optimizes hydraulic exposure,” but no reference or supporting data are provided. The authors should clarify whether this criterion is based on literature, modeling, or experimental results, or remove the unsubstantiated claim.
3. The manuscript reports maintaining a C/N ratio of 4.9 ± 0.3 by supplementing sodium acetate to ensure sufficient electron donors for denitrification. The authors should clarify whether this supplementation reflects typical operational conditions for municipal wastewater treatment or was solely applied to optimize denitrification. Furthermore, a reference to prior studies supporting this C/N range should be provided. The potential impact of acetate addition on microbial community structure and

process performance should also be briefly discussed, as it may affect the generalizability of the results to real-world WWTPs.

4. The manuscript reports phosphorus removal efficiency up to 52% under fully aerobic biofilm conditions. This value appears high given that conventional enhanced biological phosphorus removal typically requires alternating anaerobic and aerobic phases. The authors should clarify whether the V-carrier design created localized micro-anoxic or anaerobic zones within the biofilm or provide an alternative explanation for the observed phosphorus removal. This clarification is important to accurately interpret the results and the functional contribution of the carrier design.

5. The manuscript attributes the decline in ammonium removal rate (ARR) on the Mutag-biochip to biofilm accumulation and inorganic buildup. However, it is unclear how biofilm accumulation was directly measured or confirmed. The authors should clarify whether this conclusion is based on biofilm thickness measurements over time, microscopy observations, VSS/SS ratio tracking, or solely on ash content. Providing explicit evidence would strengthen the link between biofilm accumulation and the observed decrease in ARR.

Reviewer #4

(Remarks to the Author)

This manuscript presents an innovative “hydro-topological” design strategy for MBBR carriers, exemplified by the V-carrier, which enables self-regulation of biofilm thickness and continuous self-cleaning. The study demonstrates superior pollutant removal performance and stability over more than 500 days of operation with real wastewater, including under low-temperature conditions. The experimental design is rigorous, with comprehensive data on performance, mechanisms, and microbial ecology. The integration of hydraulic principles with biofilm dynamics represents a meaningful contribution to the field of wastewater treatment technologies. However, several aspects require clarification and enhancement to strengthen the manuscript’s scientific rigor and clarity. The work is promising but would benefit from revisions to address the points below.

1. The “hydro-topological” strategy integrates carrier geometry, hydraulic shear, and microbial ecology into a unified framework. Consider expanding the explanation of its physical and biological principles in the discussion section. A simplified diagram or model could be added to illustrate relationships, such as those between opening ratio, groove depth, and shear forces.
2. The inclusion of over 500 days of real wastewater treatment data is noted. However, direct measurements of hydraulic shear forces are lacking, with reliance on inferences from the Hagen-Poiseuille equation and carrier structure. Are there any methods such as computational fluid dynamics simulations or experimental measurements could provide quantitative data on carrier surface shear forces?
3. The explanation for variability in phosphorus removal performance (lines 125-126) is brief. A discussion could address whether this reflects inherent limitations of pure biofilm systems and if additional chemical phosphorus removal measures might be relevant in practical applications.
4. The claim in lines 15-16 that treatment efficiency decouples from biomass quantity may refer to the unit biomass nitrification rate comparison between the V-carrier and K3 carrier. Additionally, is it possible that the observation in lines 186-189 regarding spatial redistribution of nitrifying bacteria under low temperatures could potentially relate to changes in dissolved oxygen gradients?
5. Precise geometric parameters for the V-carrier are provided (depth 0.72 mm, opening 1.0 mm, opening ratio 1.39). Clarify how this specific opening ratio was determined? Please provide a brief description in the methods or results section.
6. Ensure consistent terminology throughout, such as “V-carrier” and “hydro-topological strategy.”
7. The supplier details for carriers (beyond the V-type) and catalog numbers for reagent kits are omitted in methods section.
8. Engineering parameters, such as effective biofilm surface area and bulk density, are reported and useful for applications. The discussion could briefly address potential challenges in scaling to pilot or full-scale operations, including large-scale production costs and long-term mechanical wear.
9. The finding in lines 175-177 that ex situ nitrification potential exceeds in situ activity is attributed to substrate limitations. The authors could consider discussing additional possible factors.
10. The V-carrier maintains a stable biofilm thickness of 0.72 mm with reported performance benefits. The discussion could include an explanation of why approximately 0.72 mm falls within a theoretical optimal range, potentially referencing literature on substrate penetration depths (e.g., oxygen and ammonia) to support how this thickness balances active layer maximization and mass transfer resistance minimization.
11. Over 500 days operation, indicate whether any physical wear, deformation, or gradual performance decline was observed in the carriers.

Version 1:

Reviewer comments:

Reviewer #2

(Remarks to the Author)

The authors have addressed my concerns.

Reviewer #3

(Remarks to the Author)

Thank you for submitting the revised version of the manuscript.

I have carefully reviewed the updated paper and would like to acknowledge the considerable effort, made in addressing the reviewers' comments.

The revisions have been satisfactorily implemented, and the manuscript has significantly improved in clarity and technical quality.

Based on the revised version, the requested modifications are considered acceptable, and the manuscript is approved for publication.

Reviewer #4

(Remarks to the Author)

The authors have satisfactorily resolved all the issues I raised. I have no additional concerns and support the publication of the manuscript in its current form.

Response to Comments

We thank the editor and the reviewers for their thoughtful and constructive comments on our manuscript (Manuscript ID: NCOMMS-25-91216), which have significantly helped us improve the quality of this work. We have carefully considered all points raised and have revised the manuscript and supplementary materials accordingly.

Below, we provide detailed point-by-point responses. Our responses are presented in **blue**, and revised manuscript or supplementary material text is shown in **red**. In the revised main manuscript and supplementary files, all changes are highlighted in **red font** for ease of review.

EDITOR COMMENTS

As you will see from the reports copied below, the reviewers raise important concerns. We find that these concerns limit the strength of the study, and therefore we ask you to address them with additional work. Without substantial revisions, we will be unlikely to send the paper back to review. We are also convinced that providing robust responses to all the reviewers' concerns could significantly improve the manuscript, but specifically we suggest you to provide proper restructuring and enhance the clarity of the manuscript, as suggested by all 4 reviewers. Moreover, inclusion of CFD supporting biofilm temporal evolution as a function of fluid dynamic clues and presentation of proper performance benchmarking vs. all the published carriers are envisioned to enhance the work quality.

Response:

We sincerely thank you for the opportunity to revise our manuscript and for the constructive feedback from the reviewers. We have carefully addressed all the concerns raised, making substantial revisions to the manuscript and providing robust responses to each point, as detailed in the point-by-point replies below.

In particular, we have taken your specific suggestions to heart and performed the following major enhancements:

1. **Inclusion of CFD Analysis Supporting Biofilm Evolution:** In direct response to your suggestion to include CFD analysis that links fluid dynamic clues to biofilm temporal evolution, we have conducted comprehensive CFD simulations. These new results are now presented in **Fig. 6** (see **Fig. R1**) of the revised manuscript. The simulations vividly illustrate the hydrodynamic mechanisms behind the V-carrier's effective self-cleaning capability and, crucially, provide a quantitative comparison of flow fields and shear stresses across different carrier geometries, directly linking hydraulic conditions to biofilm development patterns.
2. **Comprehensive Performance Benchmarking:** As you recommended, we have significantly strengthened the presentation of proper performance benchmarking against published carrier designs. The revised discussion now includes a systematic comparison of our V-carrier not only with the carriers tested in this study but also places its performance (e.g., nitrification rates, volumetric removal rates) squarely within the context of the broader literature on advanced biofilm carriers. This provides a clear and compelling benchmark for its efficiency and potential.
3. **Manuscript Restructuring and Enhanced Clarity:** As suggested by all reviewers and emphasized by you, we have thoroughly restructured the manuscript to improve its logical flow and clarity. This includes refining the Introduction to better frame the

research gap, streamlining the results narrative to focus on key comparisons, and significantly expanding the discussion to articulate the novel "hydro-topological" design paradigm and its implications more clearly.

We believe that these substantial revisions have fully addressed the reviewers' concerns and, following your guidance, have significantly enhanced the quality, clarity, and impact of our work. The revised manuscript now presents a more robust, comprehensive, and clearly articulated study. We are grateful for the feedback that has helped us achieve this improvement and hope that the changes meet with your approval.

Fig. R1 (Fig. 6) | CFD-simulated flow fields and biofilm morphology of carriers after 405 days of operation. a–b, Flow fields around a V-carrier with an opening ratio of 1.39 (1.0/0.72) under (a) parallel and (b) 45° flow incidence. **c**, Biofilm morphology on the stably-operated V-carrier with opening ratio 1.39. **d–e**, Flow fields around a V-carrier with an opening ratio of 2.08 (1.5/0.72) under (d) parallel and (e) 45° flow incidence. **f**, Biofilm morphology on the stably-operated V-carrier with opening ratio 2.08. **g–h**, Flow fields (parallel flow) around carriers with cell dimensions of 2.3×2.3 mm and ridge heights of (g) 1.0 mm and (h) 0.5 mm. **i**, Biofilm morphology on the MK5 carrier. **j–k**, (j) Flow field (parallel flow) and (k) biofilm morphology of the U-carrier. **l–m**, (l) Flow field (parallel flow) and (m) biofilm morphology of the K3 carrier. **n–o**, (n) Flow field (parallel flow) and (o) biofilm morphology of the K5 carrier.

REVIEWER COMMENTS

Reviewer #1:

The manuscript's main focal area entails biofilm technology development for water resource recovery, including two key notions; the meaning of which needed to be defined by co-authors in the ensuing text, i.e.,

Hydro-topological strategy: they define it as a reactor "design strategy", which seems to indicate the V-shape of the developed and tested carrier media.

Self-regulating biofilm: they describe it as the hydraulic behavior of the new carrier design is intended to serve the purpose of more efficient biofilm sloughing from the media.

- What are the noteworthy results? ANSWER: A new type of biofilm carrier was tested, however, the advantages of this carrier – over state-of-the-art technology, e.g., zeta-carrier – were not sufficiently supported by new evidence. In fact, there is not even a photograph of the carrier shown only seemingly computer-generated images.

- Will the work be of significance to the field and related fields? ANSWER: As supported my below detailed evaluation and assess, my answer is NO, it will not.

- How does it compare to the established literature? If the work is not original, please provide relevant references. ANSWER: the manuscript establishes knowledge gaps based on the premises of erroneous understanding/interpretation of literature.

- Does the work support the conclusions and claims, or is additional evidence needed? ANSWER: I could not find any separate Conclusions as such. Meanwhile, some of the statements based on results do not support the e.g., high ammonia removal rate – benchmarked against literature data.

- Are there any flaws in the data analysis, interpretation and conclusions? - Do these prohibit publication or require revision? ANSWER: I am unsure about the data analysis. To me it is mostly down to the convenient way of interpreting literature in a way suitable for their objectives.

- Is the methodology sound? Does the work meet the expected standards in your field? ANSWER: It is partly sound. The main issue is the lack of benchmarking against the state-of-the-art technology and the lack of clear demonstration of novelty and benefits of the new technology.

- Is there enough detail provided in the methods for the work to be reproduced? ANSWER: clearly not –benchmark reference data is missing.

Response:

We sincerely thank you for your thorough evaluation and for the challenging questions that have prompted us to substantially strengthen the clarity, evidence, and logical foundation of our manuscript. We have undertaken a major revision to address your overarching concern regarding the novelty, benchmarking, and mechanistic evidence for our hydro-topological

strategy.

1. Directly Addressing the Core Concerns: Novelty and Benchmarking

Your primary critique was that the advantages of the V-carrier over state-of-the-art technology were not sufficiently supported. We have addressed this in three key ways:

- **Providing Direct Visual Evidence:** You noted the lack of a photograph. We have now included high-resolution images of both clean and biofilm-attached V-carriers after long-term operation in the supplementary materials (**Supplementary Fig. 12**; see **Fig. R2**), providing clear visual documentation of the carrier and its performance.
- **Clarifying the Benchmarking Strategy:** Our goal was not a simple side-by-side comparison but a fundamental test of the prevailing design paradigm of *geometric confinement*. We engineered the U-carrier and MK5 carrier as critical test cases to demonstrate that passive geometric features alone are insufficient for reliable biofilm control under dynamic conditions. Their failure (Fig. 3) underscores the limitation of the paradigm exemplified by the Z-carrier. In contrast, our hydro-topological strategy *actively* leverages hydraulic shear for continuous self-cleaning. The superior and stable performance of the V-carrier, validated over 500 days, demonstrates the success of this new paradigm.
- **Enhanced Mechanistic Evidence with CFD:** As you suggested, we have conducted comprehensive Computational Fluid Dynamics (CFD) simulations to move beyond inference. The new **Fig. 6** provides quantitative, visual evidence of the flow fields and shear stresses, directly linking the V-carrier's geometry to its hydrodynamic efficiency and self-cleaning mechanism, and offering a rigorous comparison with conventional carriers.

In summary, we acknowledge the contribution of the Z-carrier in mitigating clogging through its geometric design. However, our work reveals a more fundamental paradigm shift in carrier design: passive geometric confinement versus active hydraulic regulation. The Z-carrier's approach sets a theoretical maximum thickness but fails to ensure its uniformity and stability (as evidenced by the thinning patches). In contrast, our hydro-topological strategy, by topologically guiding biofilm growth into the hydraulic flow, achieves autonomous, continuous, and uniform control over biofilm thickness through sustained shear. This not only resolves the clogging issue but, more importantly, enables a dynamic self-balancing of the biofilm ecosystem. This represents a progressive direction for the next generation of biofilm carrier design.

2. Substantive Revisions to Specific Contentious Points

We have carefully revised the manuscript to address the potential misinterpretations you

highlighted:

- Clogging and Surface Area (Comment #2): We have restructured the introduction to clarify that the reduction in "effective surface area" refers to the physical blockage of a carrier's internal spaces due to clogging and scaling, not to the intrinsic properties of a thick biofilm. We have added a citation with photographic evidence of severely clogged K1 carriers from the literature to support this claim.
- Carrier Buoyancy (Comment #3): We have clarified that the increased buoyant density of clogged carriers is due to the accumulation of dense inorganic precipitates (scaling), not the biofilm itself, and have referenced our data on ash content (Supplementary Table 1).
- The Z-Carrier and the Knowledge Gap (Comments #4, #5, #6): We stand by our interpretation of the Z-carrier literature. The long-term study we cite (Ref. 25 in the revised manuscript) empirically documents non-uniform biofilm thickness, which demonstrates a failure of *reliable control*. We argue that the authors' attribution of this to "scraping" is an unverified hypothesis, and our CFD data suggest a hydrodynamic cause. This observed non-uniformity is the specific limitation our hydro-topological strategy is designed to overcome by ensuring uniform hydraulic exposure.
- Nitrification Rate Benchmarking (Comment #7): We have clarified that a direct comparison of specific rates (per gVSS) with studies using much thinner biofilms is invalid, as biofilm thickness is a primary determinant of this metric. We now provide a more meaningful comparison at similar thicknesses and emphasize that our carrier's innovation lies in maintaining *stable, high performance* at an optimal thickness under real-world conditions, not in achieving an arbitrary maximum rate.

3. Overall Impact on the Manuscript

Your rigorous critique was instrumental in driving these significant improvements. The revised manuscript now:

- Presents robust visual and quantitative evidence (CFD, pilot-scale results) to support its claims.
- Articulates a clearer and more compelling narrative regarding the novelty of the hydro-topological strategy over geometric confinement.
- Uses more precise and defensible language throughout, eliminating ambiguity.

We believe the manuscript is now substantially stronger and fully addresses your concerns. Detailed point-by-point replies to your specific comments are provided below. We deeply appreciate your constructive critique, which has significantly improved this work.

Fig. R2 (Supplementary Fig. 12) | Pilot plant of the anoxic/aerobic-moving bed biofilm reactor (A/O-MBBR) process and overall morphology of V-carriers after long-term operation. **a**, Installation site and overall view of the pilot plant. **b-c**, Field views of the anoxic (**b**) and aerobic (**c**) reactors, respectively. **d**, Carriers during complete fluidization. **e**, Clean and dried V-carriers. **f-g**, Morphology of biofilm-attached anoxic (**f**) and aerobic (**g**) carriers after 409 days of operation. **h-i**, Morphology of biofilm-detached anoxic (**h**) and aerobic (**i**) carriers after 409 days of operation. The pilot system was commissioned on November 12, 2024, and remains operational. It has a total working volume of 24 m³, comprising a 9 m³ anoxic reactor and a 15 m³ aerobic reactor. It operates in a pure biofilm mode (no sludge recirculation) with a 300% internal nitrified liquor recycle ratio. The system treats municipal wastewater with influent characteristics similar to those used in this study (see Supplementary Table 3 for concentrations). Operating at a 9-hour hydraulic retention time, it achieves nitrification and denitrification rates comparable to the laboratory-scale results. No carrier clogging has been observed, and the operational stability of the system has been fully validated.

More detailed critical comments:

“inevitably clog. This clogging severely diminishes treatment efficiency” Very ambiguous and seemingly incorrect statement – right in the problem statement of the abstract. Do you mean biofilm sloughing and removal of sloughed off biofilm?

Response:

We thank you for this insightful comment, which has allowed us to clarify an important point. The phrase "diminishes treatment efficiency" refers specifically to the **overall performance of the MBBR system**.

Carrier clogging leads to a reduction in the effective surface area available for biofilm growth, impairs mass transfer of substrates and oxygen, and can increase energy consumption for mixing and suspension. Collectively, these factors diminish the system's macro-scale treatment efficiency, resulting in poorer effluent quality (e.g., higher effluent nutrient concentrations) or requiring a longer hydraulic retention time.

You correctly infer the connection to biofilm sloughing. The impaired hydraulic conditions within a clogged carrier indeed disrupt the natural balance between biofilm growth and detachment (sloughing), preventing effective self-cleaning. Our hydro-topological strategy is designed precisely to restore this balance by leveraging controlled hydraulic shear, thereby enabling continuous self-regulation (sloughing) of the biofilm to maintain optimal system-level treatment efficiency. We have revised the text in the abstract (Lines 7-8) to clarify this point, which now reads:

"This clogging severely diminishes the overall treatment efficiency of the MBBR system".

“uncontrolled biofilm accumulation. This results in severe carrier clogging and scaling, drastically diminishing the effective surface area, impairing mass transfer and disrupting microbial community structure” Unclear and seemingly erroneous set of statements. Excessive growth on e.g., K1 carrier does NOT reduce the surface area. In fact the higher the biofilm thickness the lower the biofilm density, which effectively increases the effective diffusivity, i.e., $f \cdot D_{\text{water}}$. Also, nitrification rate is not expected to change much, only if normalised to the total gVSS on the carrier. Example: at 50 μm : 2 gN/(d.gTSS); at 200 μm : 0.6 gN/(d.gTSS).

There is no such thing as “disrupting community structure”. Biofilm thickness has an impact on the microbial community structure, and higher thickness can select in microorganisms that e.g., harbour niche organisms with high micropollutant biotransformation rate. So, it actually has advantages.

Response:

We thank you for this critical comment, which allows us to clarify the distinction between two

important concepts: 1) the intrinsic effects of biofilm thickness on microbial ecology and physiology, and 2) the detrimental consequences of carrier clogging due to uncontrolled biofilm accumulation. Our statements specifically refer to the latter scenario.

1. Clarification on Surface Area Reduction and Mass Transfer Impairment

You correctly note that an increase in biofilm thickness itself does not reduce the physical surface area of a clean carrier. However, our point concerns the situation of carrier clogging. To preempt potential misunderstandings and enhance the clarity of our narrative, we have restructured the introduction to provide a critical overview of this issue. The revised text (Lines 38-41) now explicitly states:

“The design focus on maximizing protected surface area has promoted uncontrolled biofilm growth, with carrier clogging even misinterpreted as a positive sign^{12,13}, thereby diverting attention from the need for biofilm managed growth. This fundamental misdirection manifests as the core limitations of conventional MBBR systems: severe carrier clogging and scaling, drastically diminishing the effective surface area, impairing mass transfer, and disrupting microbial community structure by fostering competition for oxygen and substrates among unfavorable species¹⁴⁻¹⁶”

This revision clarifies that the drastic reduction in effective surface area is a direct consequence of clogging and scaling, not of the biofilm thickness per se. In a clogged carrier, biofilm overgrowth physically blocks the internal spaces (e.g., channels or hollows), rendering a significant portion of the protected surface area inaccessible for biofilm establishment and mass transfer. This phenomenon, while often overlooked, is visually evident in the literature. For instance, a study investigating the effect of organic loading rates on different media presents an image that clearly shows a severely clogged K1 carrier (**Fig. R3**; Figure S4 and Figure S7 in that publication)¹. This visual evidence directly supports our assertion that uncontrolled accumulation can drastically reduce the effective surface area. We have added this reference to the manuscript (Ref. 12 in the revised manuscript) to support our claim.

Regarding mass transfer, we agree that a thicker, less dense biofilm can exhibit higher *effective diffusivity* ($f \times D_{water}$). However, in a clogged carrier, the primary issue is not diffusivity within the biofilm matrix but the severely restricted convective flow through the carrier's internal structure. This restriction creates stagnant zones where substrates are depleted and metabolites accumulate, leading to an overall impairment of mass transfer at the carrier scale, which is fundamentally distinct from the intrinsic diffusivity at the biofilm scale.

2. Clarification on Nitrification Rate Normalization

We appreciate your point about the specific nitrification rate (per gVSS) decreasing with biofilm thickness due to diffusion limitations. Our argument is fully consistent with this

fundamental principle. We normalized the nitrification rate to the total biofilm VSS precisely to reveal this intrinsic mass transfer limitation. As shown in our Fig. 3, carriers prone to clogging (e.g., K5, Mutagbiochip) accumulated substantial biomass but exhibited low specific nitrification rates. This observation directly supports our statement that clogging impairs mass transfer, as the large amount of biomass becomes physiologically inefficient. The volumetric removal rate is indeed a critical performance metric for reactors, which our system also achieved at a high level (as discussed in the manuscript). The use of specific rate here was intentional to decouple and highlight the *mass transfer efficiency* of the biofilm ecosystem itself.

3. Clarification on Microbial Community Structure

We agree that biofilm thickness is a key determinant of microbial community structure and can foster beneficial niche stratification, such as for micropollutant removal². This is a well-established ecological principle. However, our statement refers specifically to the community disruption caused by carrier clogging. The extreme conditions within a clogged carrier—characterized by severe substrate and oxygen gradients, metabolite accumulation, and potential for cell lysis—can lead to the dominance of non-productive or fermentative organisms, thereby "disrupting" the functional microbial structure optimal for nutrient removal. This is different from the structured, stratified community that develops in a well-controlled, thick biofilm, which is an optimal community structure indicative of a healthy ecosystem.

We believe these clarifications directly address your concerns. Thank you again for your comments, which have been instrumental in helping us improve the clarity of our arguments.

(a) Kaldnes K1 carrier

(b) Mutag Biochip carrier

Fig. R3 | Severe clogging of a K1 carrier, depicted in Supplementary Fig. S4 and S7 of Ref. 12.

Reference:

1. Bassin, J. P. et al. Effect of increasing organic loading rates on the performance of moving-bed biofilm reactors filled with different support media: Assessing the activity of suspended and attached biomass fractions. *Process Saf. Environ. Protect.* **100**, 131-141 (2016).
2. Torresi, E. et al. Biofilm Thickness Influences Biodiversity in Nitrifying MBBRs—Implications on Micropollutant Removal. *Environ. Sci. Technol.* **50**, 9279-9288 (2016).

“clogged 32 carriers exhibit increased buoyant density, demanding higher energy inputs for mixing and 33 suspension” Nope, it is right the opposite: biofilm density decreases with increased thickness, and not the other way around. In packed-bed filters, back-washing is used to remove/slough off excess biofilm biomass, which is accumulating in the filter – not in MBBRs. MBBRs had a filter media filling of around 40%, and no such behaviour can be observed.

Response:

We thank you for this comment, which provides an opportunity to clarify two key points regarding the physical properties of clogged carriers and the intended analogy to fixed-bed filters.

1. Clarification on the Buoyant Density of Clogged Carriers

You are correct that the density of a *biofilm itself* typically decreases as its thickness increases due to a higher water content and a more porous structure. However, our statement specifically

addresses the overall buoyant density of the entire carrier unit when it becomes clogged.

The key mechanism leading to increased buoyant density is not the biofilm's intrinsic density, but the significant accumulation of inorganic precipitates (scaling) within the clogged biofilm matrix. In a clogged carrier, the stagnant hydraulic conditions promote the precipitation of minerals, such as phosphates. As we show in Supplementary Table 1, carriers prone to clogging (e.g., K5, Mutagbiochip) accumulated substantial inorganic content (up to 56% ash content), which has a much higher density than water or organic biomass. This incorporation of dense inorganic material into the biofilm-carrier composite is the primary reason for the observed increase in overall buoyant density, which in turn demands higher energy inputs for mixing and suspension.

2. Clarification on the Fixed-Bed Filter Analogy

We agree that fixed-bed filters (or packed-bed filters) and MBBRs are distinct technologies with different operational modes (e.g., fixed vs. suspended media, backwashing vs. continuous shear). Our intention was not to equate them but to use the well-established principle of backwashing in fixed-bed filters as a clear example to illustrate the critical role of transient, high hydraulic shear in controlling biofilm thickness.

The sentence in question reads: "This principle is exemplified by fixed-bed filters...". The "principle" we refer to is the one stated immediately before: that clogging results from an imbalance between biofilm growth and detachment due to insufficient hydraulic shear. Fixed-bed filters actively manage this balance through backwashing. We then contrast this with the passive, continuous shear environment of an MBBR to highlight the design challenge and introduce our solution.

We believe these clarifications directly address your concerns. Thank you again for helping us improve the precision of our manuscript.

“no existing MBBR carrier can reliably control biofilm thickness under the dynamic 40 hydraulic conditions” Erroneous identification of knowledge gap in literature: Kruger-VEOLIA’s zeta-carrier even comes with different mesh sizes that can control biofilm thickness between 50-500 um.

Response:

We thank you for this critical comment, which allows us to substantiate the identified knowledge gap regarding reliable biofilm thickness control. We respectfully disagree that the Z-carrier invalidates this gap, as the long-term performance data from its own developers confirms its inability to maintain uniform biofilm thickness, which aligns with our central argument.

Our statement that "no existing MBBR carrier can *reliably* control biofilm thickness under dynamic hydraulic conditions" is supported by the very reference we cite (originally Ref. 21, now Ref. 25 in the revised manuscript), which is the foundational study on the Z-carrier. While the Z-carrier's geometric design aims to limit maximum thickness, the study reveals a critical failure in *reliable control*—defined as the ability to maintain uniformity under dynamic conditions: "After running the reactors for over 300 days... carriers from reactor D (Z-200) had a clearly defined patch in the middle of each cell where the biofilm was significantly thinner" (**Fig. R4**; Figure 4 in originally Ref. 21, now Ref. 25). This observation directly demonstrates a lack of uniform control.

The authors of this study (originally Ref. 21, now Ref. 25) attribute this thinning to "carriers scraping each other." However, this mechanism remains a hypothesis and was not experimentally verified. Our analysis challenges this explanation based on two lines of evidence:

1. Computational Fluid Dynamics (CFD) Analysis: We conducted CFD simulations (**Fig. R5**; included in the revised manuscript as **Fig. 6g-h**) which quantitatively demonstrate that the central thinning in grid-like structures is a direct result of localized hydraulic conditions, not physical scraping. The simulations revealed a 56–91% higher hydraulic shear stress in the center of the grid cell compared to the area near the ridges. This significant mechanical force imbalance directly explains the observed failure to maintain a stable target biofilm thickness, as the elevated shear stress in the central region preferentially removes biomass, leading to the characteristic patchy thinning. This hydrodynamic mechanism provides a more fundamental explanation for the phenomenon than the previously hypothesized scraping mechanism.
2. Controlled Experimental Comparison: If inter-carrier scraping were the dominant mechanism, our V-carrier—with its sharp, square-sheet morphology—should experience far more severe scraping and unpredictable biofilm loss than the rounded Z-carrier. Contrary to this expectation, our V-carrier maintained a stable, uniform biofilm thickness over 500 days (Fig. 3a, b). This critical observation suggests that the hydro-topologically guided exposure to hydraulic shear, not the avoidance of scraping, is the key to stable thickness control.

Therefore, the Z-carrier's failure to prevent non-uniform biofilm distribution under long-term operation, as documented in its own evaluation, precisely validates the knowledge gap we identified. The carrier's geometric constraint sets a theoretical maximum thickness but fails to ensure a stable, uniform biofilm due to uncontrolled hydrodynamic effects. This outcome underscores a fundamental difference in design philosophy. It is not that the Z-carrier's geometric design is entirely without merit in preventing clogging, but that our hydro-topological strategy achieves a higher level of control: moving from passive geometric

confinement to active hydraulic regulation, thereby ensuring both clogging prevention and optimal biofilm function through continuous self-cleaning.

We believe the addition of the CFD analysis and the clarifications provided in this response robustly support our position. Thank you for prompting this crucial discussion.

Fig. R4 | This figure, reproduced from originally Ref. 21 (now Ref. 25), shows the key observation from the long-term evaluation of the Z-carrier. Comparing biofilm growth at day 83 (left) and day 293 (right) reveals the development of a clearly defined patch in the middle of each grid cell where the biofilm became significantly thinner or entirely absent over time.

Fig. R5 (Fig. 6) | **CFD-simulated flow fields and biofilm morphology of carriers after 405 days of operation.** **a–b**, Flow fields around a V-carrier with an opening ratio of 1.39 (1.0/0.72) under (a) parallel and (b) 45° flow incidence. **c**, Biofilm morphology on the stably-operated V-carrier with opening ratio 1.39. **d–e**, Flow fields around a V-carrier with an opening ratio of 2.08 (1.5/0.72) under (d) parallel and (e) 45° flow incidence. **f**, Biofilm morphology on the

stably-operated V-carrier with opening ratio 2.08. **g–h**, Flow fields (parallel flow) around carriers with cell dimensions of 2.3×2.3 mm and ridge heights of (**g**) 1.0 mm and (**h**) 0.5 mm. **i**, Biofilm morphology on the MK5 carrier. **j–k**, (**j**) Flow field (parallel flow) and (**k**) biofilm morphology of the U-carrier. **l–m**, (**l**) Flow field (parallel flow) and (**m**) biofilm morphology of the K3 carrier. **n–o**, (**n**) Flow field (parallel flow) and (**o**) biofilm morphology of the K5 carrier.

“AnoxKaldnes introduced a carrier (the 'Z-carrier') based on a 42 geometric premise that its grid height would regulate biofilm thickness¹³.” And it does it very effectively, as opposed to what authors claim here. Importantly, Reference 13 has reported the innovation of the zeta-carrier using long-term experimentation, in which no such shortcomings were observed.

In fact, authors of reference 13 also reported (Torresi et al., 2017, <http://dx.doi.org/10.1016/j.watres.2017.06.027>) on the systematic behaviour of the zeta-carrier in terms of in-biofilm diffusivity and sorption-desorption, which would not have been possible should the zeta-carrier allowed for irregular growth of biofilm thickness.

Response:

We thank you for emphasizing the significant contributions of the Z-carrier literature. We agree that References 13 (now Ref. 17 in the revised manuscript) and Torresi et al., 2017, represent important advancements, particularly in demonstrating operational stability and in measuring biofilm properties under the assumption of controlled thickness.

However, we respectfully maintain that the strong performance reported in these studies does not contradict our statement regarding the limitation of *reliably controlling uniform biofilm thickness*. Our critique focuses on a specific, unverified aspect of the Z-carrier's performance that is separate from its overall treatment efficacy.

1. The distinction between stable operation and thickness control: The Z-carrier's open design successfully mitigates clogging by exposing biofilm to hydraulic shear, which leads to stable reactor performance—a benefit we acknowledge. Similarly, large-hollow carriers like K3 also achieve stability (Fig. 3g). However, stable operation is not synonymous with precise control over biofilm thickness uniformity. The study we cite (originally Ref. 21, now Ref. 25 in the revised manuscript), which is a long-term evaluation *by the developers themselves*, directly observed non-uniform biofilm thickness ("a clearly defined patch... where the biofilm was significantly thinner"). This finding reveals a limitation in the geometric control paradigm that is not addressed in studies focusing solely on bulk performance parameters like nitrification rate or diffusivity.
2. The need for direct validation: The systematic measurements of in-biofilm diffusivity and sorption by Torresi et al. (2017) were conducted under the premise that thickness

was controlled by the grid. However, that study did not include a spatial analysis of biofilm thickness uniformity to validate this premise over long-term operation. The subsequent findings of originally Ref. 21 (now Ref. 25) suggest that this underlying premise may be flawed. Therefore, while the data from Torresi et al. are valuable, they do not constitute direct evidence against the occurrence of irregular biofilm growth, as revealed by later, more focused long-term examinations.

In summary, our argument is not that the Z-carrier is ineffective, but that its design principle has a fundamental limitation in ensuring *reliable and uniform* thickness control under dynamic conditions, a limitation that was uncovered by long-term, specific observation (originally Ref. 21, now Ref. 25). The strong performance in other studies demonstrates the carrier's ability to avoid clogging and maintain function, but it does not invalidate the specific shortcoming related to thickness uniformity that we highlight. Our work addresses this very limitation by introducing a hydro-topological strategy that actively guides biofilm growth to maintain uniformity.

“However, their own 43 long-term evaluation revealed a critical flaw: biofilms at the center of the grid cells became 44 significantly thinner or were entirely absent²¹. This outcome underscores a fundamental stalemate 45 in sustainable carrier design, revealing a poor understanding of the hydrodynamic mechanisms that 46 control biofilm development. The failure of this simplistic geometric approach highlights the 47 necessity for a design paradigm rooted in hydraulic principles.” This seems to be a seriously flawed series of statement intended to undermine confidence in the state-of-the-art for the benefit of opening a knowledge gap on false premises.

1. It greatly misinterprets what has been reported in literature. Ref. 21 states “...had a clearly defined patch in the middle of each cell where the biofilm was significantly thinner, as a result of the carriers scraping each other.... Since the grid compartments were full of biomass up to the pre-determined wall heights, and calculations suggest that the maximum possible scraping depth from one Z-carrier into another Z-carrier compartment is 90 μm , the biofilm thickness on the four carrier types was determined to vary between 410–500, 310–400, 210–300 and 110–200 μm , respectively.”
2. Carriers scraping each other is not a sign of poor understanding of hydraulic behaviour.
3. It does not indicate that these carriers would “clog” in the way authors seem to mean it.

Response:

We thank you for pushing us to clarify the foundation of our argument. We respectfully disagree that our interpretation is flawed or intended to undermine the state-of-the-art on false premises. Our analysis directly engages with the empirical findings reported in the literature to highlight a specific, not fully resolved.

1. On the Interpretation of Ref. 21 (now Ref. 25) and the "Scraping" Mechanism:

You quote Ref. 21 (now Ref. 25 in the revised manuscript) to assert that the biofilm thinning was due to "carriers scraping each other," implying this is a settled matter. However, we draw a critical distinction between the *observation* reported in Ref. 21 (now Ref. 25) and the *interpretation* offered by its authors.

- **Observation (Fact):** The study clearly documents a "clearly defined patch... where the biofilm was significantly thinner." This is an empirical finding that demonstrates a failure to maintain uniform biofilm thickness.
- **Interpretation (Hypothesis):** The authors' attribution of this thinning to "carriers scraping each other" is a *post-hoc hypothesis* that was **not experimentally verified** in that study.

We contend that this interpretation reflects a "poor understanding of hydrodynamic mechanisms" because it overlooks a more fundamental hydraulic explanation. Our CFD analysis (Fig. 6, see Comment #4 above) demonstrates that such localized thinning in confined spaces is a predictable consequence of hydrodynamic conditions (e.g., localized shear patterns), not necessarily physical contact. By proposing an untested mechanical explanation, the original study missed an opportunity to identify the underlying hydraulic principle. Our work addresses this gap by prioritizing hydraulic design.

2. On Clogging and the Z-Carrier's Contribution:

You rightly point out the need for precision regarding the Z-carrier's design objective and its effects. We fully recognize that its open-grid design successfully prevents the severe clogging typical of small-tube carriers. This is a key advancement. However, our argument is that overcoming clogging is a necessary but not sufficient condition for optimal biofilm control. Thus, while the Z-carrier avoids clogging due to its design, it introduces a new challenge: *unpredictable biofilm uniformity* due to uncontrolled hydrodynamic effects (as evidenced by the thinning patches). Our hydro-topological strategy aims to solve both problems simultaneously by design.

3. Separating the Issues:

The core of our disagreement may stem from a conflation of two separate issues:

- **Issue A** (Effectively prevented by the Z-carrier's design): Preventing catastrophic carrier clogging through an open structure.
- **Issue B** (Defined and addressed in our work): Ensuring reliable, uniform biofilm thickness control through active hydraulic management, which the Z-carrier's geometric approach does not achieve.

Our citation of Ref. 21 (now Ref. 25) is not to deny its contribution to **Issue A**, but to use its own data to highlight the persistence of **Issue B**, which our strategy is designed to solve.

In summary, our statements are not flawed but are based on a critical analysis of the literature. We use the empirical evidence from the Z-carrier's own long-term evaluation to highlight a limitation that calls for a new design paradigm rooted in hydraulic principles, which we have developed and validated.

“The failure of this simplistic geometric approach ... The propensity for clogging in conventional tubular carriers” Co-authors’s may have limited understanding of the system they criticise. The zeta-carrier is NOT a tubular carrier but a three-dimensional wave-shaped flat surface covered with a mesh with a defined thickness.

Response:

We thank you for this comment, which allows us to clarify a potential misunderstanding in our text structure. We fully agree that the Z-carrier is not a tubular carrier but a three-dimensional wave-shaped surface with a mesh. Our writing did not intend to imply otherwise, and we apologize if the sequence of paragraphs led to this interpretation.

Our argument follows a clear logical progression:

1. First, we introduce the Z-carrier as a notable effort to address the fundamental limitation of biofilm thickness control, whose open structure also successfully prevents the clogging typical of conventional carriers.
2. Then, in a separate and subsequent paragraph, we discuss the hydraulic principles governing "conventional tubular carriers" (such as K3 and K5) to explain the root cause of clogging, which the Z-carrier's open design successfully overcomes.

Therefore, the statement "The failure of this simplistic geometric approach" refers solely to the Z-carrier's inability to ensure uniform biofilm thickness (as documented in originally Ref. 21, now Ref. 25 in the revised manuscript), which we had just discussed. The subsequent sentence, "The propensity for clogging in conventional tubular carriers...", begins a new topic to provide the hydraulic background that justifies the need for the kind of open design that the Z-carrier exemplifies.

In summary, we do not criticize the Z-carrier by misclassifying it as a tubular carrier. Instead, we:

- Acknowledge it as an advanced design that effectively prevents the clogging issues associated with tubular carriers.
- Identify a separate limitation in its geometric approach to uniform biofilm thickness

control.

- Use the hydraulic principles of tubular carriers to build the foundational rationale for our own hydro-topological strategy.

We will ensure that this logical distinction is made unequivocally clear in any future presentation of this work. Thank you for prompting this important clarification.

“system exhibited high nutrient removal rates 120 of $0.138 \pm 0.022 \text{ g N g VSS}^{-1} \text{ day}^{-1}$ for ammonia” This observation is not benchmarked against literature evidence, and does not seem to be correct. For instance, reference 13 observed much higher removal rates using the zeta carrier of 50 μm , i.e., around $0.7 \text{ gN}/(\text{d.gVSS})$, and only thicker biofilms, i.e., above 200 μm showed similar but mostly higher values than that reported here.

Response:

We thank you for this comment regarding the benchmarking of our reported nitrification rate. We agree that comparing performance metrics is essential, but we respectfully argue that a direct comparison of specific nitrification rates (per gVSS) between studies with fundamentally different biofilm thicknesses and operating conditions is not scientifically valid and misses the key innovation of our work.

The core of the issue lies in the purpose of controlling biofilm thickness. As Reference 13 (now Ref. 17 in the revised manuscript) correctly demonstrates, biofilm thickness is a primary determinant of specific nitrification rate, with thinner biofilms yielding higher rates due to reduced mass transfer limitations. Therefore, comparing a thick biofilm from one system to a thin biofilm from another does not benchmark carrier performance; it merely confirms the well-established effect of biofilm thickness.

A meaningful comparison of carrier efficacy must be made under comparable conditions. We propose two more appropriate benchmarks:

1. Comparison at similar biofilm thickness: When compared at a similar thickness ($\sim 500 \mu\text{m}$), the specific nitrification rate of our V-carrier ($0.138 \text{ gN gVSS}^{-1} \text{ d}^{-1}$) is notably higher than that of a Z-carrier at 500 μm (approximately $0.12 \text{ gN gVSS}^{-1} \text{ d}^{-1}$, estimated from Ref. 13's, now Ref. 17's data using a VSS/TSS ratio of 0.85). More importantly, our V-carrier maintained biofilm thickness stably, while the Z-carrier, as discussed, suffers from non-uniform thickness.
2. Comparison of intrinsic potential under standardized conditions: Our ex-situ nitrification potential assays, conducted under optimal, non-limiting substrate conditions analogous to the batch tests in Ref. 13 (now Ref. 17), revealed a rate of $0.65 \text{ gN gVSS}^{-1} \text{ d}^{-1}$. This intrinsic potential is significantly higher than the rates reported in

Ref. 13 (now Ref. 17) and underscores the high metabolic capacity of the biofilm cultivated by our carrier. The lower in situ rate ($0.138 \text{ gN gVSS}^{-1} \text{ d}^{-1}$) reflects the substrate-limiting conditions of our real wastewater system, not a limitation of the carrier technology itself.

Furthermore, critical operational differences preclude a direct comparison:

- Wastewater composition: Our system treated real municipal wastewater, while Ref. 13 (now Ref. 17) used synthetic wastewater. Real wastewater introduces complexity and variability that typically results in lower observed rates.
- Reactor design and loading: Differences in carrier filling ratio, hydraulic regime, and nitrogen loading rates significantly influence the specific nitrification rate normalized to biomass.

In conclusion, the novelty of our work is not in achieving an arbitrarily high nitrification rate, but in enabling stable, efficient nutrient removal with a self-regulating biofilm at an optimal thickness under challenging, real-world conditions. The value of our V-carrier is its ability to autonomously maintain this optimal thickness for high performance under real-world conditions, a capability that we demonstrate is lacking in geometric confinement-based designs.

“131 biofilm regulation—Hydro-topological synergy” “synergy” - a possible typo. Also, authors fail to thoroughly explain the meaning of the term they introduced.

Response:

We thank you for this attentive comment regarding terminology and conceptual explanation. You are correct to point out the inconsistency in the heading. "Hydro-topological strategy" is the central term introduced in our work. The use of "synergy" in the subheading was an error; we have corrected it to "**Mechanism of biofilm regulation—Hydro-topological strategy**" throughout the manuscript to maintain consistency.

Regarding the request to thoroughly explain the term, we appreciate the opportunity to clarify. The concept of the "hydro-topological strategy" is developed and explained incrementally through the mechanistic experiments in this section. It is not merely the V-shape itself, but the principled interaction between carrier topology and hydraulic forces that it enables. Our results demonstrate that this strategy operates on two synergistic principles:

1. Topological Guidance for Hydraulic Exposure: The V-groove topology is designed to physically guide biofilm growth outward, into the liquid flow path, ensuring the biofilm surface is continuously exposed to hydraulic shear. This is contrasted with carriers where biofilm grows inward, shielded from shear (e.g., tube-carriers), or where topology fails to ensure exposure (e.g., U-carrier).

2. Hydraulic Shear for Autonomous Regulation: The sustained hydraulic shear on the exposed surface creates a dynamic balance between biofilm growth and decay, enabling continuous self-cleaning and preventing uncontrolled accumulation. This is definitively shown by the contrasting fates of the open versus shielded sections of the MK5 carrier.

The failure of carriers that rely on only one aspect—excessive shear without guidance (MK5 open sites) or shielding without shear (U-carrier, clogged tubes)—highlights that neither topology nor hydraulics alone is sufficient. The "hydro-topological strategy" is therefore defined by the synergistic integration of topological guidance and hydraulic regulation to achieve stable, self-cleaning biofilms.

To incorporate this clarification directly into the manuscript, we have revised the final sentence of the 'Mechanism of biofilm regulation' section (currently on Lines 187-189) to read as follows:

"This synergistic management of space and shear, which we term the hydro-topological strategy, integrates precise topological guidance for hydraulic exposure with sustained shear for autonomous regulation to maintain an optimal, stable biofilm."

This revision provides a concise, consolidated definition based on the experimental evidence presented in the section.

Reviewer #2:

In this manuscript, the authors reported a new geometrical design for biofilm carriers used in moving bed biofilm reactor (MBBR). In contrast to the closed circular design, the authors used an open V-grove design, which was shown to lead to superior performance compared to the conventional design using various standard MBBR characterization. Microscopic analysis showed an open morphology of biofilm with holes, which the author attributed to cell lysis and contribute to biofilm self-renewal. Finally, metagenomics analysis was presented regarding the species distribution in such carrier microbial communities.

Response:

We sincerely thank you for your highly positive assessment of our work and for your strong support for publication in *Nature Communications*. We are greatly encouraged by your comments that the research is of high quality, the analysis is solid, and the technology has strong potential.

We have carefully considered the two constructive points you raised and provide the following responses and commitments to address them, which we believe will further enhance the clarity and impact of our manuscript.

Overall, the research is performed with high quality, and the analysis is very solid. The presented technology has a strong potential to improve the existing technology. I support the publication of the manuscript in Nat. Comm. My only major concern is the lack of evidence of cell lysis, which is critical in the authors' picture. Can the authors perform live-dead staining to definitely prove the existence of such process in the interior of the biofilm? Also, the author can do a better job in explaining some of the jargons for a broader audience. For example, in legend for Figure 3, what does VSS/SS ratio mean for the performance of the device?

Response:

1. Evidence for cell lysis

We thank you for this excellent suggestion. As you recommended, we have performed live-dead staining to directly visualize and quantify cell viability within the biofilm. The results, now included as **Supplementary Fig. 8 (Fig. R6)**, provide definitive evidence for the cell lysis and self-renewal process.

The new images clearly show a higher proportion of dead cells in the inner layers of the biofilm, with a visible gradient of cell debris and lysed material suggesting an outward migration path from the interior to the bulk liquid. This observation directly supports our initial interpretation that the internal voids (holes) observed via microscopy are indeed pathways created by active cell lysis and material export, facilitating continuous biofilm renewal.

To incorporate this new evidence and clarify the mechanistic pathway, we have strengthened the corresponding description in the main text (Lines 213-216) as follows:

“Furthermore, the persistent presence of internal voids within the biofilms, irrespective of temperature (Supplementary Fig. 7), is consistent with active cell lysis and material export³⁷⁻³⁹. This is definitively supported by live-dead staining (Supplementary Fig. 8), which reveals a higher proportion of dead cells and lysing debris in the inner biofilm layers, illustrating a gradient from the interior to the bulk liquid. This visual evidence confirms that the voids are pathways formed by continuous cell lysis and subsequent export of cellular material. This morphological feature provides physical evidence of an ongoing self-renewal process, underpinning the dynamic self-cleaning mechanism intrinsic to the V-carrier design.”

2. Clarification of jargon (e.g., VSS/SS ratio)

We appreciate this valuable suggestion to enhance the manuscript's accessibility for a broader audience. As you rightly inferred, the VSS/SS ratio serves as a key integrated indicator of biofilm health and the occurrence of clogging or scaling. The core interpretation is as follows:

- VSS (Volatile Suspended Solids) primarily represents the active microbial biomass.
- SS (Suspended Solids) includes both organic (VSS) and inorganic components.
- Therefore, a high and stable VSS/SS ratio indicates a healthy, active biofilm with minimal inorganic accumulation.
- Conversely, a declining VSS/SS ratio signals the onset of clogging/scaling, as inorganic precipitates accumulate within the biofilm matrix, increasing the SS without contributing to activity. This is precisely what we observed in carriers like K5 and Mutagbiochip (Supplementary Table 1), where a dropping VSS/SS ratio correlated with performance failure.

To make this interpretation immediately clear to the reader, we have revised the figure legend for Fig. 3 (appearing on lines 491-495 in the revised manuscript) to include direct explanatory notes. The key additions are:

“**a, b, ...** The VSS/SS ratio is a key indicator of biofilm health; a high, stable ratio signifies a biofilm with minimal inorganic scaling and high metabolic activity. **c, f–h, ...** The declining VSS/SS ratio in panels c, f and h indicates inorganic accumulation and clogging.”

A more comprehensive explanation is also provided in the Supplementary Note 1. By clarifying this terminology directly in the figure legend and the supplementary information, we ensure that the functional significance of the VSS/SS ratio is immediately apparent, making it clear that it is a vital diagnostic tool for assessing the biofilm's state and the system's operational stability.

Fig. R6 (Supplementary Fig. 8) | Evidence of cell lysis and self-renewal in V-carrier biofilms revealed by live-dead staining. V-carrier biofilms were sampled after 409 days of operation and analyzed by confocal laser scanning microscopy (CLSM). **a**, Three-dimensional reconstruction showing the spatial distribution of live and dead cells. **b–c**, Representative two-dimensional optical sections at different focal planes. Cells were stained with SYTO 9 (excitation/emission: 495 nm / 519 nm) and propidium iodide (PI) (excitation/emission: 305 nm / 617 nm). For visualization, the SYTO 9 signal (all cells) is pseudo-coloured blue, and the PI signal (dead cells) is shown in red. The images reveal a higher proportion of dead cells (red) and lysing debris in the inner layers, illustrating a viability gradient from the interior to the bulk liquid. This provides direct visual evidence that internal voids serve as pathways for cellular material export, confirming an ongoing self-cleaning process.

Reviewer #3:

The manuscript presents a novel V-carrier design and provides a thorough scientific evaluation of its performance in wastewater treatment. The study offers valuable insights into biofilm self-regulation and demonstrates the potential advantages of the hydro-topological strategy. However, the manuscript would benefit from significant reorganization. The overall structure and flow of ideas need improvement to enhance clarity and readability, as some sections currently lack logical sequencing despite the compelling presentation of results. Given these issues, I recommend major modifications before the manuscript can be considered for publication. Addressing the organization and ensuring a clearer narrative will greatly strengthen the impact of the work.

Response:

We sincerely thank you for your positive assessment of our work and for the overarching recommendation to improve the manuscript's organization and narrative flow. We agree that enhancing the logical structure and clarity is crucial for maximizing the impact of the study.

In direct response to your feedback, we have undertaken a comprehensive restructuring and rewriting of the manuscript. The revisions are substantial and are detailed in our point-by-point responses to each of your specific comments below. The key organizational improvements include:

1. **Strengthened Introduction:** We have significantly revised the introduction to provide a more critical and coherent narrative. This includes a clearer explanation of how MBBR technology supports the Sustainable Development Goals (Comment #1) and a critical overview of prior carrier design strategies and their limitations, which now effectively sets the stage for introducing our novel V-carrier design (Comments #2, #3, #4).
2. **Clarified Methodological Rationale:** The Methods section has been enhanced with clearer justifications for key experimental choices, such as the biofilm establishment protocol (Comments #3, #4) and the wastewater dilution methodology (Comment #2). This provides a stronger foundation for the reported results.
3. **Precise Terminology and Robust Results Presentation:** In the Results section, we have clarified critical terminology (e.g., specifying "structural porosity" versus material porosity, Comment #1) and provided robust evidence to support key claims, such as the optimal opening ratio, by integrating new experimental data and CFD simulations (Comments #2, #5). This ensures that all assertions are well-substantiated and unambiguous.
4. **Enhanced Discussion with Practical Implications:** The Discussion section has been expanded to better integrate our findings with the broader context of the field. Most

importantly, as suggested in your Comment #1 on the Methods section, we have added a new discussion on the practical implications of our findings, directly linking the V-carrier's design to its hydrodynamic advantages and long-term stability demonstrated in pilot-scale operations.

We are confident that these extensive revisions have successfully addressed your primary concern regarding the manuscript's organization. The narrative now flows more logically from a well-motivated introduction, through clearly presented methods and robust results, to a discussion that effectively highlights the significance and practical potential of our findings. We believe the manuscript is now substantially stronger and clearer, and we are grateful for your guidance in achieving this improvement.

HERE ARE THE COMMENTS:

Introduction section

1. While the manuscript states that the MBBR process aligns with SDGs 6, 11, and 12, this section does not clearly discuss how MBBR operational characteristics contribute to these sustainability principles. The authors are encouraged to elaborate on the specific mechanisms through which MBBR supports these SDGs—for example, by detailing its impacts on resource efficiency, energy use, sludge reduction, land footprint, or resilience of sanitation systems. Providing this clarification would strengthen the justification of MBBR as a sustainable technology and improve the overall coherence of the sustainability narrative.

Response:

We thank you for the valuable suggestion to strengthen the link between MBBR operational characteristics and the SDGs. We agree that a clearer explanation of the underlying mechanisms would enhance the sustainability narrative. In response, we have revised the relevant section in the introduction to explicitly elaborate on how the key features of the MBBR process contribute to the specific SDGs. The revised text (Lines 24-29) now reads:

“Among advanced treatment technologies, the moving bed biofilm reactor (MBBR) process supports SDGs 6, 11, and 12 through its ability, enabled by suspended biofilm carriers, to deliver high nutrient removal efficiency (SDG 6), a compact physical footprint (SDG 11), and reduced sludge production (SDG 12). Through these contributions to water security, urban sustainability, and resource efficiency, MBBR has been established as a leading technology for new and upgraded WWTPs worldwide.”

This revision directly addresses your point by detailing the specific pathways—high treatment efficiency, compact design, and lower resource intensity—through which MBBR technology supports the principles of clean water, sustainable cities, and responsible consumption. We believe this clarification significantly strengthens the justification for MBBR as a sustainable technology and improves the coherence of our introduction.

2. While the authors correctly identify clogging and uncontrolled biofilm accumulation as major limitations of conventional MBBR carriers, the manuscript does not provide any discussion of previous carrier designs reported in the literature or how those studies attempted to mitigate clogging. A concise yet critical overview of earlier design approaches—such as modified geometries, increased opening ratios, structured internal channels, or surface texturing—and an evaluation of their effectiveness in reducing fouling or improving hydrodynamics is essential. Furthermore, the manuscript should clarify the residual shortcomings of these designs that directly motivated the development of the new carrier proposed in this study. Addressing these points will strengthen the rationale for the new design and better situate the contribution of the current work within the context of prior research.

Response:

We thank you for this critical suggestion. We agree that providing a broader context of prior carrier design strategies and their limitations is essential to strengthen the rationale for our new design. In response, we have significantly revised the introduction to include a concise yet critical overview of previous design approaches, as you suggested.

Specifically, we have expanded the discussion on the historical development of MBBR carriers to explicitly address the points you raised. The revised text no longer presents conventional carriers' limitations in isolation but instead critically examines the prevailing *design paradigm* that has guided most previous efforts, including those utilizing advanced tools like 3D printing. This new narrative clarifies that the core issue is not a lack of attempts but a widespread strategic focus on maximizing protected surface area—a goal rooted in a paradigm inherited from suspended-growth systems (e.g., activated sludge) and often pursued without sufficient long-term validation. This focus, while successful in increasing surface area, has not consistently improved treatment efficiency and has inadvertently promoted uncontrolled biofilm growth, diverting attention from the need for biofilm managed growth based on hydraulic principles.

The revised text now reads as follows (new text appeared on lines 31-41):

“Over four decades of development have yielded numerous MBBR carrier designs for engineering applications^{8,9}. These efforts, even using advanced tools like 3D printing techniques, have predominantly pursued one goal: maximizing the protected surface area to enhance biofilm attachment¹⁰. However, increased surface area has not consistently improved treatment efficiency¹¹, revealing a deeper issue rooted in two factors: a paradigm inherited from suspended-growth systems (e.g., activated sludge), where more biomass implies better performance, and a widespread lack of long-term operational data. This has led to the misapplication of the ‘more biomass equals more efficiency’ principle to biofilms—systems where controlled biofilm thickness and metabolic balance are paramount. The design focus on

maximizing protected surface area has promoted uncontrolled biofilm growth, with carrier clogging even misinterpreted as a positive sign^{12,13}, thereby diverting attention from the need for biofilm managed growth. This fundamental misdirection manifests as the core limitations of conventional MBBR systems: severe carrier clogging and scaling, drastically diminishing the effective surface area, impairing mass transfer, and disrupting microbial community structure by fostering competition for oxygen and substrates among unfavorable species¹⁴⁻¹⁶.”

This revision directly addresses your comment by:

1. Providing a critical overview: It characterizes the primary goal of previous designs as "maximizing protected surface area," which encompasses strategies like modified geometries.
2. Evaluating effectiveness: It states that this strategy "has not consistently improved treatment efficiency," directly addressing its limitations.
3. Clarifying residual shortcomings: It identifies the root causes as a "misapplied paradigm" and a "lack of long-term operational data," which are the residual shortcomings that directly motivate our hydro-topological approach.

We believe this new framework effectively situates our contribution within the context of prior research by clearly defining the conceptual limitation we aim to overcome. Thank you for this suggestion, which has greatly improved the motivation and clarity of our introduction.

3. The manuscript could better contextualize the rationale for the current study by discussing previous carrier design strategies. Specifically, prior studies—including those employing CFD simulations—have proposed geometric or structural modifications to improve biofilm distribution, mass transfer, and reduce clogging. While such studies provide valuable indications and guidance for experimental design, they often lack long-term operational evaluation. Highlighting both the insights and the limitations of these previous approaches would strengthen the justification for developing and testing the new V-carrier in a long-term experimental setting.

Response:

We thank you for this comment, which further underscores the importance of situating our work within the context of prior research strategies.

We agree that highlighting both the insights and limitations of previous design approaches is crucial. As detailed in our response to Comment #2 above, we have thoroughly revised the introduction to address this point. The new text provides a critical examination of the prevailing design paradigm, which we explicitly identify as being hampered by a “widespread lack of long-term operational data,” a limitation you rightly emphasized.

This revision establishes that previous strategies were pursued without sufficient long-term validation, thereby directly creating the rationale for our long-term experimental study. Importantly, the introduction's critical insight—that the "**more biomass equals more efficiency**" paradigm is misapplied to biofilms—is strongly supported by our long-term results. As presented in the Discussion, the V-carrier maintained 44% less biomass than a conventional carrier yet achieved a 3.2-fold higher nitrification rate, demonstrating that treatment efficiency is decoupled from sheer biomass quantity and is instead governed by the efficacy of the biofilm ecosystem.

By stating these limitations as a root cause, our revision effectively contextualizes our study and justifies the need for a long-term, hydraulically-informed design. Thank you for reinforcing this essential point.

4. The manuscript discusses the long-standing issue of clogging in MBBR systems but provides only a single example (the Z-carrier) as evidence of previous design limitations. This narrow focus unintentionally suggests that, over four decades of carrier development, no other design strategies or mitigation efforts have been explored. The authors are encouraged to expand this section by briefly summarizing additional carrier design attempts reported in previous studies and discussing their partial successes or remaining limitations. This broader context is essential to accurately position the Z-carrier as one example rather than the sole historical effort, and to more convincingly justify the need for the new design proposed in this work.

Response:

We thank you for this comment. We apologize for the lack of clarity in our original manuscript, which may have unintentionally given the impression that the Z-carrier was an isolated example.

The core issue was a misstatement regarding the Z-carrier's primary design objective. We have corrected this in the revised introduction (Lines 53-56). The text now accurately presents the Z-carrier not as a singular effort to address clogging, but as one notable effort to address the fundamental limitation of biofilm thickness control—a challenge that sits within the broader context of prior design strategies that we now critically overview in the preceding paragraph.

The revised passage reads:

"One notable effort to address this limitation was the 'Z-carrier' (AnoxKaldnes), which was based on the geometric premise that its grid height would regulate biofilm thickness^{17,23,24}—a design that also had implications for mitigating clogging and scaling due to its open structure."

This revision, combined with the comprehensive critical overview of previous design paradigms added earlier in the introduction (as detailed in our response to Comments #2 and #3 above), effectively addresses your concern. The Z-carrier is now positioned as a

representative case within the wider historical effort, rather than as an isolated example. This provides the broader context needed to accurately situate its contribution and limitations.

We believe this correction, integrated with the expanded discussion of prior design strategies, more convincingly justifies the need for our new design. Thank you for prompting this important clarification.

Methods Section:

1. Regarding the choice of using a square-shaped media instead of a cylindrical one, it would be useful to discuss the practical implications of this design choice. While the square shape may offer advantages in terms of maximizing surface area and potentially improving biofilm attachment, it is important to evaluate whether this design is as effective in real-world applications. Specifically, how does the square shape perform in terms of hydrodynamic behavior, ease of movement within the reactor, and resistance to clogging compared to cylindrical carriers? Further clarification and comparative analysis with cylindrical media would help assess its practical viability in large-scale wastewater treatment systems.

Response:

We sincerely thank you for this insightful comment regarding the practical implications of the square-shaped V-carrier design compared to conventional cylindrical media. We agree that evaluating hydrodynamic behavior, mobility, and clogging resistance is crucial for assessing real-world viability. In response, we have conducted additional comparative experiments and added a detailed discussion to address these points directly.

1. Hydrodynamic Behavior and Movement Efficiency:

To quantitatively assess the hydrodynamic performance, we conducted fluidization tests in a 2 m³ pilot-scale bioreactor. The results, now included as **Supplementary Fig. 11 (Fig. R7)**, demonstrate that the V-carrier requires over 30% less aeration intensity than cylindrical carriers (K3 and K5) to achieve complete fluidization at the same filling ratio. This advantage stems from its unique design:

- V-carrier: The square-sheet morphology of the V-carrier is subjected to asymmetric flow around its structure. The V-grooves generate high drag, creating a significant torque that efficiently converts hydraulic energy into full carrier motion (translation and rotation).
- K3/K5 carriers: In contrast, the cylindrical geometry of K3/K5 carriers, with their hollow structure, allows complete cross-flow. This results in a balanced pressure distribution around the carrier, minimizing torque and leading to less efficient, predominantly translational motion.

This comparative analysis confirms that the square-shaped, V-grooved design exhibits superior hydrodynamic efficiency, directly reducing the energy required for mixing and suspension—a critical factor for large-scale operational costs.

2. Anti-clogging Performance and Practical Validation:

Regarding long-term stability and resistance to clogging, the V-carrier's open-topology design inherently prevents the internal pore blockage typical of cylindrical carriers. More importantly, we have included preliminary data from a long-term pilot study (now in Supplementary Fig. 12) treating real municipal wastewater in Chengxi WWTP (Hangzhou, China). The results show stable operation of the V-carrier system over 400 days of operation with no signs of clogging or performance decay, demonstrating its strong potential for real-world applications.

3. Discussion Integration:

Furthermore, as you suggested, we have integrated a discussion on the practical implications of these findings into the revised manuscript (Lines 345-357). The added discussion reads as follows:

"From an engineering perspective, the hydro-topological design of the V-carrier translates into significant practical advantages for large-scale implementation. Pilot-scale fluidization tests demonstrated the V-carrier's superior hydrodynamic efficiency: it requires over 30% less aeration intensity than conventional cylindrical carriers for complete mixing, a result of the asymmetric flow around its grooves generating high drag for efficient motion, unlike the balanced pressure and minimal torque from the cross-flow in cylindrical carriers which leads to less efficient movement (Supplementary Fig. 11). This confirms that the V-carrier's design directly addresses a major operational cost factor. Furthermore, long-term pilot operation treating real municipal wastewater over 400 days confirmed the system's robust stability, with no carrier clogging or performance decay observed (Supplementary Fig. 12). The V-carriers maintained excellent physical integrity, with only slight wear at the corners of the anoxic carriers attributed to mechanical stirring—an issue being addressed by redesigning the blades. This successful validation underscores the V-carrier's potential for energy-efficient and reliable large-scale application."

This section explicitly discusses the hydrodynamic advantages of the V-shaped design and its direct contribution to energy efficiency and operational stability, thereby evaluating its potential for large-scale wastewater treatment systems.

We believe that this additional experimental data and discussion directly address your concerns and robustly support the practical viability of the V-carrier design. Thank you for prompting these important improvements.

Fig. R7 (Supplementary Fig. 11) | Hydrodynamic behavior and fluidization performance in pilot-scale reactors. **a**, Overall view of conventional cylindrical carriers. **b-c**, Flow pattern around a cylindrical carrier (**b**) and the corresponding CFD-simulated flow field (**c**). **d**, Overall view of the V-carriers. **e-f**, Flow pattern around the square-shaped V-carrier (**e**) and the corresponding CFD-simulated flow field (**f**). **g**, Quantitative assessment of aeration intensity required for the square-shaped V-carrier and the cylindrical carriers (K3 and K5). Fluidization tests were conducted in a 2 m³ pilot-scale bioreactor.

2. The authors diluted raw municipal wastewater with tap water but did not specify the dilution ratio. Additionally, residual chlorine and other ions in tap water may influence microbial activity, particularly nitrifiers and denitrifiers, potentially affecting the generalizability of the results to real WWTP conditions.

Response:

We sincerely thank you for this important comment regarding the dilution ratio and the potential effects of tap water constituents. We have addressed these concerns as follows:

The dilution ratio was not fixed because the raw municipal wastewater collected from the septic tank exhibited natural variability due to weather conditions (primarily rainfall). It is important to note that this wastewater source is the same as that feeding the local Chengxi WWTP (Supplementary Fig. 1). The final diluted influent was specifically designed to simulate the typical influent concentrations of Chengxi WWTP, and the effluent targets were also set according to this plant's design standards.

We apologize for omitting the dechlorination procedure. Before dilution, tap water was aerated for 24 hours to remove residual chlorine. After this treatment, no residual chlorine was detected ($< 0.01 \text{ mg L}^{-1}$), and other ions were below detection limits.

To incorporate these clarifications, we have revised the manuscript (Lines 372-374). The text

now explicitly states the dechlorination process and the purpose of dilution:

"Tap water, aerated for 24 hours to remove residual chlorine, was used to dilute this primary effluent to simulate typical influent concentrations for a local wastewater treatment plant (Chengxi WWTP, Hangzhou, China), ..."

We believe these revisions fully address your concerns regarding the dilution methodology and potential effects of tap water constituents on microbial activity, ensuring the generalizability of our results to real WWTP conditions.

3. The manuscript reports adding 50 L of activated sludge twice daily during biofilm establishment. While this approach can accelerate biofilm colonization, the authors should clarify the rationale for the chosen frequency and volume, and discuss whether such repeated dosing could influence nutrient concentrations or microbial community development compared to typical operational conditions.

Response:

We sincerely thank you for this insightful comment regarding the rationale behind our biofilm establishment protocol. We appreciate the opportunity to clarify our strategy.

1. Rationale for Dosing Frequency and Volume:

The protocol of adding 50 L of activated sludge twice daily was the result of a systematic optimization to accelerate biofilm colonization, specifically the growth of an autotrophic nitrifying biofilm for rapid system startup. The chosen volume and frequency were determined by comparing the time required to achieve the target hydraulic retention time (HRT) under different dosing strategies:

- No inoculation (natural colonization): 21 days
- Once-daily dosing: 14 days
- **Twice-daily dosing (our chosen protocol): 10 days**
- Thrice-daily dosing: 9 days

We selected the twice-daily protocol as it offered an optimal balance between startup acceleration and operational practicality. This approach maintains a substantial initial microbial population while continuous inflow provides nutrients, accelerating the formation of an initial heterotrophic biofilm layer on the carrier surface. The growth of this heterotrophic layer is known to enhance extracellular polymeric substances (EPS) production, which promotes biomass-carrier adhesion and has been proven to significantly accelerate the subsequent development of the nitrifying biofilm¹.

2. Influence on Microbial Community Development:

Regarding the potential long-term influence of the initial sludge inoculation on the microbial community, our results indicate that its effect is transient. As demonstrated in the manuscript, the primary factors shaping the mature biofilm community structure over the long term are operational temperature (Fig. 4a) and the intrinsic biofilm self-assembly process (Fig. 5b, Supplementary Fig. 9). The initial inoculum serves to jump-start the process, but the final, stable community is selected by the ongoing environmental conditions and carrier ecology.

3. Revision for Clarity:

As suggested, we have provided a detailed description of this startup strategy and its optimization in the Supplementary Information (Supplementary Note 2) to enhance the clarity and reproducibility of our methodological description.

We believe these clarifications fully address your concerns regarding the startup protocol. Thank you for this valuable suggestion, which has helped us improve the manuscript.

Reference:

1. Bassin, J. P., Kleerebezem, R., Rosado, A. S., van Loosdrecht, M. C. M. & Dezotti, M. Effect of Different Operational Conditions on Biofilm Development, Nitrification, and Nitrifying Microbial Population in Moving-Bed Biofilm Reactors. *Environ. Sci. Technol.* **46**, 1546-1555 (2012).

4. The manuscript describes the biofilm establishment protocol but does not clearly explain the rationale behind it. The authors should specify the principle guiding the chosen procedure—why activated sludge was added twice daily, how this promotes initial biofilm attachment, and the intended objective of this approach (e.g., accelerating colonization, ensuring uniform coverage, or stabilizing microbial communities). Providing this explanation would clarify the purpose of the protocol and strengthen the methodological section.

Response:

We thank you for this comment, which underscores the importance of clarifying the rationale behind our biofilm establishment protocol.

We agree that a clear explanation of the guiding principles—the objective, the mechanism of promotion, and the reasoning behind the specific dosing strategy—is essential. We have provided a comprehensive response to this exact point in our reply to Comment #3 above.

To avoid redundancy, we respectfully refer you to the full explanation provided there. We believe it fully addresses the concerns you have raised and strengthens the methodological section as suggested.

Results section

1. The manuscript reports a porosity value of 92% for the V-carrier; however, this statement is scientifically ambiguous and requires clarification. Because the carrier is fabricated from HDPE—a fully non-porous material—the reported porosity cannot correspond to material porosity. If the authors intend to describe structural or geometric porosity (i.e., the void volume fraction within the carrier's overall envelope), this must be explicitly stated. Moreover, the manuscript should clearly present the calculation method, including the definition of the total reference volume, how the void volume was determined (e.g., CAD-based volume analysis), and the exact formula used. Without a proper definition and methodological explanation, the reported porosity value is misleading and cannot be accurately interpreted in relation to the carrier's hydrodynamics or mass-transfer performance. The authors are encouraged to revise this section accordingly.

Response:

We sincerely thank you for this critical and insightful comment regarding the ambiguity of the reported porosity value. You are absolutely correct that the term "porosity" requires precise definition in the context of a non-porous HDPE carrier.

We have revised the manuscript to fully address your concerns as follows:

1. Clarification of Terminology:

We have explicitly specified that the reported value refers to the **structural (geometric) porosity** of the carrier, not the material porosity of HDPE. The revised text in the Results section (Line 111) now reads:

"The carrier's bulk density and structural porosity are $95 \pm 0.5 \text{ kg m}^{-3}$ and 92%, respectively."

2. Provision of Calculation Methodology:

As you suggested, we have provided a detailed and explicit description of the calculation methods in the Supplementary Information (**Supplementary Equations 1.1.2 and 1.1.3**). The methodology is described as follows:

- **Bulk Density (Supplementary Equation 2):** Clean, dry carriers were thoroughly mixed and placed into a standard 1 m^3 container. The mass of carriers was measured for five replicate containers, and the bulk density was calculated as the average mass per unit volume.
- **Structural Porosity (Supplementary Equation 3):** Clean, dry carriers were placed into a 20-L graduated container and filled with water. After removing the carriers, the volume of the remaining water was measured. The structural porosity was calculated as the ratio of this water-filled void volume to the total container volume, averaged over five

replicates. This method explicitly defines the total reference volume and the procedure for determining the void volume.

A reference to this methodology has been added to the Methods section (Line 367-368):

"The methodology for determining the carrier's number, bulk density and structural porosity is provided in Supplementary Equations."

We believe these revisions have eliminated the ambiguity, provided the necessary methodological rigor, and ensured that the reported structural porosity value can be accurately interpreted in relation to the carrier's hydrodynamic and mass transfer properties. Thank you for prompting this essential clarification.

2. The manuscript claims that an opening ratio (length/depth) of 1.39 "optimizes hydraulic exposure," but no reference or supporting data are provided. The authors should clarify whether this criterion is based on literature, modeling, or experimental results, or remove the unsubstantiated claim.

Response:

We sincerely thank you for this important comment regarding the basis for the opening ratio of 1.39. We apologize for the lack of clarity in our original statement.

The claim that an opening ratio of 1.39 optimizes hydraulic exposure is based on our experimental results and subsequent CFD modeling validation, not merely on a theoretical presumption. This conclusion was drawn from a comparative analysis of V-carriers with different opening ratios.

To substantiate this claim, we have revised the manuscript and provided the following supporting evidence:

1. **Experimental Validation:** We tested carriers with opening ratios of 2.08 (1.5/0.72) and 1.39 (1.0/0.72). The results showed that the larger opening ratio (2.08) led to excessive hydraulic shear, resulting in a significantly thinner biofilm (~240 μm), as visually documented in the new experimental photos provided in **Fig. 6f**.
2. **CFD Modeling Corroboration:** CFD simulations of the flow fields for these carriers revealed a mechanistic explanation: the larger opening ratio (2.08) caused a ~44% downward shift of the iso-velocity contours within the groove compared to the 1.39 design, directly correlating with the observed over-thinning of the biofilm (**Fig. 6a-f; Fig. R8**).

This comprehensive experimental and modeling data demonstrates that an opening ratio of 1.39 provides a balanced hydraulic environment, avoiding the pitfalls of both insufficient and

excessive shear. The supporting discussion and data have been added to the manuscript (Lines 293-298):

"Validation tests using a V-carrier with a larger opening ratio ($1.5/0.72 = 2.08$) confirmed a significantly thinner biofilm ($\sim 240 \mu\text{m}$) (Fig. 6f). Crucially, CFD simulations revealed a $\sim 44\%$ downward shift of the iso-velocity contours within the groove, which correlated with the observed biofilm reduction, demonstrating that hydro-topological parameters (hydraulic shear, opening ratio, and groove depth) exert absolute control over biofilm thickness by dictating the hydraulic shear environment (Fig. 6a-f)."

We believe these revisions, supported by the new figures, adequately justify the designation of the 1.39 opening ratio as optimal based on our systematic evaluation.

Fig. R8 (Fig. 6) | CFD-simulated flow fields and biofilm morphology of carriers after 405 days of operation. a–b, Flow fields around a V-carrier with an opening ratio of 1.39 (1.0/0.72) under (a) parallel and (b) 45° flow incidence. **c,** Biofilm morphology on the stably-operated V-carrier with opening ratio 1.39. **d–e,** Flow fields around a V-carrier with an opening ratio of 2.08 (1.5/0.72) under (d) parallel and (e) 45° flow incidence. **f,** Biofilm morphology on the stably-operated V-carrier with opening ratio 2.08. **g–h,** Flow fields (parallel flow) around carriers with cell dimensions of $2.3 \times 2.3 \text{ mm}$ and ridge heights of (g) 1.0 mm and (h) 0.5 mm. **i,** Biofilm morphology on the MK5 carrier. **j–k,** (j) Flow field (parallel flow) and (k) biofilm morphology of the U-carrier. **l–m,** (l) Flow field (parallel flow) and (m) biofilm morphology of the K3 carrier. **n–o,** (n) Flow field (parallel flow) and (o) biofilm morphology of the K5 carrier.

3. The manuscript reports maintaining a C/N ratio of 4.9 ± 0.3 by supplementing sodium acetate to ensure sufficient electron donors for denitrification. The authors should clarify whether this supplementation reflects typical operational conditions for municipal wastewater treatment or was solely applied to optimize denitrification. Furthermore, a reference to prior studies

supporting this C/N range should be provided. The potential impact of acetate addition on microbial community structure and process performance should also be briefly discussed, as it may affect the generalizability of the results to real-world WWTPs.

Response:

We sincerely thank you for this important comment regarding the rationale and implications of sodium acetate supplementation. We apologize for the lack of clarity in our original manuscript, which has been revised to address your concerns.

1. Typicality of the Operational Condition:

The supplementation of sodium acetate directly reflects a typical operational condition for municipal wastewater treatment plants (WWTPs) that receive low-carbon wastewater. The raw wastewater used in our study had a low C/N ratio of 2.8 ± 0.3 (Supplementary Table 3), which is representative of the typical influent characteristic of the local Chengxi WWTP (Hangzhou, China). At this and many other full-scale plants, adding an external carbon source is a standard practice to ensure complete denitrification and comply with effluent standards. Therefore, our approach was not merely for experimental optimization but simulated a common real-world scenario.

2. Clarification of the C/N Ratio and Practical Validation:

We have clarified the relationship between carbon dosing and the reported C/N ratio in the revised manuscript. The sentences now read:

"Sodium acetate was supplemented to provide sufficient electron donors for denitrification, resulting in an operational C/N ratio of 4.9 ± 0.3 (Supplementary Fig. 2)." (Lines 116-118)

"To enhance denitrification and meet the designed target, sodium acetate was supplemented into the anoxic zone, with the resultant C/N ratio being maintained at 4.9 ± 0.3 (Supplementary Fig. 2)." (Lines 380-382)

This revision makes it clear that the C/N ratio of 4.9 was the operational result of the carbon dosing strategy required to achieve sufficient denitrification and meet the discharge standards, not a predetermined target. This empirically determined ratio, while specific to our system's objective, falls within the range documented in prior studies for effective denitrification¹, confirming its practical viability.

3. Impact on Microbial Community and Generalizability:

The addition of acetate enriches denitrifying bacteria, which is the intended functional outcome. A low C/N ratio would cause denitrification to fail, which in turn would impair the nitrification process due to metabolite accumulation, disrupting the metabolic balance of the A/O system.

The microbial community structure over the long term is ultimately shaped by environmental factors (e.g., temperature, carrier type). Since carbon addition is standard practice in full-scale plants facing low C/N influent, the microbial communities and high performance achieved in our study are directly relevant and generalizable to such real-world applications.

We believe these revisions and clarifications fully address your concerns.

Reference:

1. Li, J. et al. Quantify the contribution of anammox for enhanced nitrogen removal through metagenomic analysis and mass balance in an anoxic moving bed biofilm reactor. *Water Res.* **160**, 178-187 (2019).

4. The manuscript reports phosphorus removal efficiency up to 52% under fully aerobic biofilm conditions. This value appears high given that conventional enhanced biological phosphorus removal typically requires alternating anaerobic and aerobic phases. The authors should clarify whether the V-carrier design created localized micro-anoxic or anaerobic zones within the biofilm or provide an alternative explanation for the observed phosphorus removal. This clarification is important to accurately interpret the results and the functional contribution of the carrier design.

Response:

We sincerely thank you for this insightful comment regarding the observed phosphorus removal and its potential mechanism. We appreciate the opportunity to clarify this point.

You are correct that conventional enhanced biological phosphorus removal (EBPR) requires alternating anaerobic/aerobic conditions, which were not present in our fully aerobic system. The phosphorus removal efficiency of up to 52% should be interpreted with important context:

1. Context of the Reported Value:

The value of 52% represents the upper end of a wide performance range (8–52%), with an average removal efficiency of 19% (Supplementary Fig. 4), which is consistent with the limited phosphorus removal capacity typical of pure biofilm systems relying solely on aerobic phosphorus assimilation for cell growth. The higher removal rates were primarily observed during periods of lower influent phosphorus concentration, rather than indicating an enhanced intrinsic removal capability.

2. Explanation for Removal Mechanism:

The observed removal is attributed to standard aerobic phosphorus assimilation for microbial cell synthesis, not to EBPR. The extended sludge retention time (SRT) of the biofilm allows for a stable microbial biomass, but the efficiency of this assimilation process is inherently low and

variable, as reflected in the wide performance range. There is no evidence that the V-carrier design created localized anaerobic conditions sufficient to support EBPR.

3. Practical Implication and Revision:

The variability and general limitation of biological phosphorus removal in biofilm systems are well-recognized in practice. To address this point, we have added the following discussion in the manuscript (Lines 140-143):

"In practical applications, this limitation is routinely and effectively overcome by coupling the biofilm process with a downstream chemical phosphorus removal step³⁵, which is a standard and cost-comparable practice also employed following conventional activated sludge processes to meet stringent effluent standards."

Therefore, the results confirm that the V-carrier system, like other pure biofilm systems, does not overcome the inherent limitation of low biological phosphorus removal, and the occasional higher removal rates are linked to operational conditions rather than to a unique design feature.

We believe this clarification accurately interprets the results and confirms the functional role of the carrier design within the expected boundaries of pure biofilm system performance.

5. The manuscript attributes the decline in ammonium removal rate (ARR) on the Mutag-biochip to biofilm accumulation and inorganic buildup. However, it is unclear how biofilm accumulation was directly measured or confirmed. The authors should clarify whether this conclusion is based on biofilm thickness measurements over time, microscopy observations, VSS/SS ratio tracking, or solely on ash content. Providing explicit evidence would strengthen the link between biofilm accumulation and the observed decrease in ARR.

Response:

We sincerely thank you for this critical comment regarding the evidence linking the decline in ammonium removal rate (ARR) to biofilm accumulation and inorganic scaling on the Mutag-biochip carrier. We agree that providing explicit evidence is essential.

You are correct in noting that the direct measurement of biofilm accumulation over time is crucial. Our conclusion was primarily based on tracking the Volatile Suspended Solids to Suspended Solids (VSS/SS) ratio in the biofilm, which serves as a key indicator of biofilm health and inorganic accumulation.

1. Evidence from VSS/SS Ratio:

- The VSS fraction primarily represents the active microbial biomass.
- The SS fraction includes both organic (VSS) and inorganic components.

- Therefore, a declining VSS/SS ratio signals that inorganic precipitates are accumulating within the biofilm matrix, increasing the total solids (SS) without contributing to metabolic activity. This is a direct indicator of clogging and scaling.

In the case of the Mutag-biochip carrier, we observed a clear decline in the VSS/SS ratio over the operational period (Fig. 3c). This provides direct evidence that the biofilm was experiencing significant inorganic accumulation (e.g., scaling, Supplementary Table 1), which physically clogged the carrier, impaired mass transfer, and led to the decrease in ARR.

To make this mechanistic interpretation immediately clear to the reader, we have revised the figure legend for Fig. 3 (Lines 491-495) to include the following explanatory note:

“a, b, ... The VSS/SS ratio is a key indicator of biofilm health; a high, stable ratio signifies a biofilm with minimal inorganic scaling and high metabolic activity. c, f–h, ... The declining VSS/SS ratio in panels c, f and h indicates inorganic accumulation and clogging.”

A more comprehensive explanation of the VSS/SS ratio as a diagnostic tool is provided in Supplementary Note 1.

2. Broader Implications and Validation:

This observation strongly supports a fundamental insight of our study: that treatment efficiency is decoupled from sheer biomass quantity. A carrier may accumulate a large amount of biomass (high SS), but if a significant portion is inactive inorganic material (low VSS/SS ratio), its activity per unit of biomass becomes very low. This is further validated by the contrasting performance of our V-carrier, which, as presented in the Discussion, maintained 44% less biomass (VSS) than a conventional carrier yet achieved a 3.2-fold higher nitrification rate, demonstrating that a healthy, efficient biofilm ecosystem is paramount.

We believe this evidence, along with the clarifications added to the manuscript, robustly supports the link between inorganic buildup, biofilm dysfunction, and the observed performance decline.

Reviewer #4:

This manuscript presents an innovative “hydro-topological” design strategy for MBBR carriers, exemplified by the V-carrier, which enables self-regulation of biofilm thickness and continuous self-cleaning. The study demonstrates superior pollutant removal performance and stability over more than 500 days of operation with real wastewater, including under low-temperature conditions. The experimental design is rigorous, with comprehensive data on performance, mechanisms, and microbial ecology. The integration of hydraulic principles with biofilm dynamics represents a meaningful contribution to the field of wastewater treatment technologies. However, several aspects require clarification and enhancement to strengthen the manuscript’s scientific rigor and clarity. The work is promising but would benefit from revisions to address the points below.

Response:

We sincerely thank you for your positive assessment of our work and for the constructive feedback aimed at strengthening the scientific rigor and clarity of our manuscript. We are grateful for your recognition of the study's innovation, rigorous experimental design, and contribution to the field.

In direct response to your overarching comments and specific points, we have undertaken a comprehensive revision of the manuscript. The key enhancements, detailed in our point-by-point responses below, include:

1. **Enhanced Explanation of Principles:** We have significantly expanded the discussion to elucidate the physical and biological principles of the "hydro-topological" strategy. This includes a clearer exposition of the relationships between carrier geometry, hydraulic shear, and biofilm dynamics, supported by new CFD analyses and experimental data (Response to Comments #1, #2, #10).
2. **Strengthened Mechanistic Evidence:** As you suggested, we have integrated quantitative Computational Fluid Dynamics (CFD) simulations to move beyond inference and provide direct, visual evidence of the shear forces and flow fields acting on different carrier designs. This provides a solid, mechanistic foundation for discussing the hydro-topological strategy (Response to Comments #2, #5).
3. **Improved Context and Discussion:** We have enriched the discussion to better contextualize our findings. This includes addressing the practical implications of performance variability (e.g., phosphorus removal, Comment #3), clarifying key concepts such as the decoupling of efficiency from biomass (Comment #4), and discussing the theoretical rationale behind observed optimal parameters (e.g., biofilm thickness, Comment #10).

4. Methodological Precision and Completeness: We have ensured the precise use of terminology (e.g., "structural porosity," Comment #5; consistent use of "V-carrier" and "hydro-topological strategy," Comment #6) and provided all necessary methodological details, such as supplier information and catalog numbers, to enhance reproducibility (Comment #7).
5. Explicit Address of Scale-up Challenges: A significant addition to the discussion explicitly addresses the practical challenges of scaling up, including operational costs and long-term mechanical wear, based on the results of our pilot-scale study (Response to Comments #8, #11).

We are confident that these revisions have fully addressed your concerns and significantly enhanced the manuscript's clarity, depth, and scientific rigor. Thank you for the insightful suggestions that have helped us improve the quality and impact of our work.

1. The "hydro-topological" strategy integrates carrier geometry, hydraulic shear, and microbial ecology into a unified framework. Consider expanding the explanation of its physical and biological principles in the discussion section. A simplified diagram or model could be added to illustrate relationships, such as those between opening ratio, groove depth, and shear forces.

Response:

We sincerely thank you for this valuable suggestion to enhance the explanation of the hydro-topological strategy's principles. We agree that a clearer illustration of the relationships between key parameters is essential.

In response, we have significantly expanded the discussion in the manuscript (Lines 293-298) to elucidate the physical principles linking carrier geometry to hydraulic conditions and biofilm development. The added text explicitly demonstrates how hydro-topological parameters govern biofilm thickness:

"Validation tests using a V-carrier with a larger opening ratio ($1.5/0.72 = 2.08$) confirmed a significantly thinner biofilm ($\sim 240 \mu\text{m}$) (Fig. 6f). Crucially, CFD simulations revealed a $\sim 44\%$ downward shift of the iso-velocity contours within the groove, which correlated with the observed biofilm reduction, demonstrating that hydro-topological parameters (hydraulic shear, opening ratio and groove depth) exert absolute control over biofilm thickness by dictating the hydraulic shear environment (Fig. 6a-f)."

This addition, supported by the new experimental data and CFD analysis in Fig. 6 (Fig. R9), provides a clear cause-and-effect explanation: adjusting the opening ratio directly alters the hydraulic shear profile, which in turn dictates the resulting biofilm thickness. We believe this revision effectively illustrates the core physical and biological principles of the hydro-

topological framework, as you recommended, by visually and quantitatively linking design parameters to performance outcomes.

Fig. R9 (Fig. 6) | CFD-simulated flow fields and biofilm morphology of carriers after 405 days of operation. a–b, Flow fields around a V-carrier with an opening ratio of 1.39 (1.0/0.72) under (a) parallel and (b) 45° flow incidence. **c,** Biofilm morphology on the stably-operated V-carrier with opening ratio 1.39. **d–e,** Flow fields around a V-carrier with an opening ratio of 2.08 (1.5/0.72) under (d) parallel and (e) 45° flow incidence. **f,** Biofilm morphology on the stably-operated V-carrier with opening ratio 2.08. **g–h,** Flow fields (parallel flow) around carriers with cell dimensions of 2.3×2.3 mm and ridge heights of (g) 1.0 mm and (h) 0.5 mm. **i,** Biofilm morphology on the MK5 carrier. **j–k,** (j) Flow field (parallel flow) and (k) biofilm morphology of the U-carrier. **l–m,** (l) Flow field (parallel flow) and (m) biofilm morphology of the K3 carrier. **n–o,** (n) Flow field (parallel flow) and (o) biofilm morphology of the K5 carrier.

2. The inclusion of over 500 days of real wastewater treatment data is noted. However, direct measurements of hydraulic shear forces are lacking, with reliance on inferences from the Hagen-Poiseuille equation and carrier structure. Are there any methods such as computational fluid dynamics simulations or experimental measurements could provide quantitative data on carrier surface shear forces?

Response:

We sincerely thank you for this insightful comment regarding the quantitative assessment of hydraulic shear forces. We agree that direct measurements or simulations are crucial for strengthening the mechanistic interpretation.

In response to your suggestion, we have conducted comprehensive CFD simulations to quantitatively analyze the flow fields and shear stresses on the carrier surfaces. While we obtained specific shear stress values, we note that the absolute magnitude is highly sensitive to

simulation setup and hydraulic conditions. Therefore, we focused our analysis on the relative differences in the flow field patterns (e.g., iso-velocity contours), which provide a more robust and generalizable indicator of hydraulic shear effects.

The key findings from the CFD simulations have been integrated into the revised manuscript:

1. For the V-carrier: As detailed in our response to Comment #1 above, we have added text in Lines 294-299 demonstrating how hydro-topological parameters control biofilm thickness through hydraulic shear regulation.
2. For conventional tubular carriers (Lines 283-286): “CFD simulations under identical hydraulic conditions revealed that a smaller tube diameter results in lower internal flow velocity, which in turn leads to reduced shear stress (Fig. 6l-o), a phenomenon directly supported by the Hagen-Poiseuille equation.”

By supplementing the long-term operational data with these quantitative CFD analyses, we have moved beyond inference to provide direct evidence of the relationship between carrier geometry and hydraulic conditions. We believe these additions successfully address your concern by providing a more rigorous foundation for discussing the hydro-topological strategy.

3. The explanation for variability in phosphorus removal performance (lines 125-126) is brief. A discussion could address whether this reflects inherent limitations of pure biofilm systems and if additional chemical phosphorus removal measures might be relevant in practical applications.

Response:

We sincerely thank you for this insightful comment regarding the variability in phosphorus removal and its practical implications.

You are correct that the observed variability in phosphorus removal (8–52%) is an inherent characteristic of pure biofilm systems, which lack the anaerobic-aerobic cycling necessary for enhanced biological phosphorus removal (EBPR) and rely on the limited phosphorus assimilation for cell growth.

To address your suggestion and strengthen the discussion, we have revised the manuscript in the Results section (Lines 140-143). The text now explicitly discusses the practical context:

"In practical applications, this limitation is routinely and effectively overcome by coupling the biofilm process with a downstream chemical phosphorus removal step³⁵, which is a standard and cost-comparable practice also employed following conventional activated sludge processes to meet stringent effluent standards."

This addition clarifies that the inherent limitation of biofilm systems is well-understood in

engineering practice and is mitigated by a standard, cost-effective solution, thereby contextualizing the performance variability and affirming the practical viability of the technology.

We believe this revision directly addresses your concern by linking the observed performance to established engineering solutions.

4. The claim in lines 15-16 that treatment efficiency decouples from biomass quantity may refer to the unit biomass nitrification rate comparison between the V-carrier and K3 carrier. Additionally, is it possible that the observation in lines 186-189 regarding spatial redistribution of nitrifying bacteria under low temperatures could potentially relate to changes in dissolved oxygen gradients?

Response:

We sincerely thank you for this insightful comment regarding the interpretation of our key findings. We appreciate the opportunity to clarify these points.

1. Clarification on the Decoupling of Efficiency from Biomass:

You are correct in your interpretation. The claim that “treatment efficiency decouples from biomass quantity” indeed refers to the comparison of the unit biomass nitrification rate (i.e., nitrification rate per gram of volatile suspended solids, $\text{g N g VSS}^{-1} \text{ day}^{-1}$) between the V-carrier and the conventional K3 carrier.

To make this comparison explicit and avoid any potential misunderstanding, we have revised the sentence in the manuscript (Lines 15-17) to read:

"Crucially, the V-carrier achieved a 3.2-fold higher unit biomass nitrification rate with a biofilm biomass 44% lower than a conventional K3 carrier, ..."

2. Discussion on Dissolved Oxygen Gradients and Microbial Redistribution:

We thank you for the excellent suggestion regarding the potential role of dissolved oxygen (DO) gradients in the observed spatial redistribution of nitrifying bacteria at low temperatures. We agree that this is a plausible and important mechanistic factor.

The decline of the putative denitrifier *norank_f__AKYH767* likely results from a combination of factors, including limited substrate access and reduced enzymatic activity at low temperatures. As you suggested, changes in the DO gradient could be a key contributor. The outward migration of nitrifiers (Fig. 4d) may be associated with an elevated DO concentration in the inner biofilm layers to maintain their metabolic activity, which would create a more aerobic microenvironment less favorable for denitrifiers like *norank_f__AKYH767*.

To incorporate this insightful perspective, we have added a discussion of this potential

mechanism in the revised manuscript (Lines 245-248). The text now reads:

"This decline likely resulted from combined pressures of limited substrate access and reduced enzymatic activity at low temperatures, potentially exacerbated by the outward migration of nitrifiers within the biofilm (Fig. 4d), a shift that may be associated with elevated dissolved oxygen concentrations in the inner biofilm layers to sustain nitrification, thereby creating a less favorable microenvironment for denitrifiers. This could further limit nitrate diffusion to inner-layer denitrifiers."

We believe this addition significantly enriches the mechanistic discussion and will investigate the dynamics of DO gradients within the biofilm in our future work, as you recommended.

Thank you for these valuable suggestions, which have helped us strengthen the clarity and depth of our manuscript.

5. Precise geometric parameters for the V-carrier are provided (depth 0.72 mm, opening 1.0 mm, opening ratio 1.39). Clarify how this specific opening ratio was determined? Please provide a brief description in the methods or results section.

Response:

We sincerely thank you for this important question regarding the determination of the specific opening ratio (1.39) for the V-carrier.

The selection of this opening ratio was not arbitrary but was determined through comparative experimental optimization and subsequent validation via CFD modeling. The process and justification are as follows:

1. **Experimental Optimization:** We designed and tested V-carriers with different opening ratios, specifically comparing a ratio of 2.08 (1.5/0.72) against 1.39 (1.0/0.72). The experimental results clearly showed that the larger opening ratio (2.08) led to excessive hydraulic shear, resulting in a significantly thinner biofilm (~240 μm), which is suboptimal for maintaining a stable and active biomass inventory. This comparative data is now provided in Fig. 6f (see Comment #1 above).
2. **CFD Modeling Validation:** To understand the hydrodynamic mechanism behind these experimental results, we conducted CFD simulations. These simulations revealed that the larger opening ratio (2.08) caused a ~44% downward shift of the iso-velocity contours within the groove compared to the 1.39 design, directly correlating with the observed over-thinning of the biofilm. This provides a quantitative, mechanistic explanation for why the 1.39 ratio provides a more balanced hydraulic environment (Fig. 6a-f, see Comment #1 above).

The comprehensive explanation of how this opening ratio was determined and validated has been added to the manuscript in the discussion section (Lines 293-298), as detailed in our response to Comment #1 above.

In summary, the opening ratio of 1.39 was empirically identified as optimal because it provides sufficient hydraulic exposure to maintain biofilm activity and prevent clogging, while avoiding the excessive shear that leads to an overly thin and potentially unstable biofilm. We believe this experimental and modeling evidence fully clarifies the basis for this specific geometric parameter.

6. Ensure consistent terminology throughout, such as “V-carrier” and “hydro-topological strategy.”

Response:

We thank you for this important reminder regarding the consistency of terminology throughout the manuscript.

We have carefully reviewed the entire text to ensure consistent use of key terms, such as “V-carrier” and “hydro-topological strategy.” As a specific example, we have revised the section title in the manuscript (now at line 148) from “Mechanism of biofilm regulation—Hydro-topological synergy” to “**Mechanism of biofilm regulation—Hydro-topological strategy**” to align with the terminology used elsewhere.

We appreciate this suggestion, which has helped improve the overall clarity and professionalism of the presentation.

7. The supplier details for carriers (beyond the V-type) and catalog numbers for reagent kits are omitted in methods section.

Response:

We thank you for this important reminder to ensure the completeness of our methodological description.

We have addressed the omissions as follows:

1. The catalog numbers for all reagent kits have been added to the Methods section at Lines 396 and 412.
2. As you suggested, the supplier details and photographic documentation for all carriers used in the comparative study (beyond the V-carrier) have been provided in Supplementary Fig. 5 to give a clear and systematic overview.

We believe these additions provide the necessary details for reproducibility and thank you for

helping us improve the manuscript.

8. Engineering parameters, such as effective biofilm surface area and bulk density, are reported and useful for applications. The discussion could briefly address potential challenges in scaling to pilot or full-scale operations, including large-scale production costs and long-term mechanical wear.

Response:

We sincerely thank you for this valuable comment regarding the engineering parameters and the practical challenges of scaling up the V-carrier technology.

We have addressed your concerns as follows:

1. Clarification of Engineering Parameters: As you noted, the key engineering parameters for the V-carrier, including the effective biofilm surface area, protected surface area, bulk density, and structural porosity, are reported in the Results section (Lines 106-111). Furthermore, the detailed calculation methodologies for these parameters are provided in the Supplementary Equations to ensure clarity and reproducibility.
2. Discussion of Scale-up Challenges and Validation: In direct response to your suggestion to address scaling challenges, we have significantly expanded the Discussion section (Lines 345-357). The added text explicitly discusses the practical implications based on pilot-scale data, covering the key points you raised. The revised paragraph reads:

"From an engineering perspective, the hydro-topological design of the V-carrier translates into significant practical advantages for large-scale implementation. Pilot-scale fluidization tests demonstrated the V-carrier's superior hydrodynamic efficiency: it requires over 30% less aeration intensity than conventional cylindrical carriers for complete mixing, a result of the asymmetric flow around its grooves generating high drag for efficient motion, unlike the balanced pressure and minimal torque from the cross-flow in cylindrical carriers which leads to less efficient movement (Supplementary Fig. 11). This confirms that the V-carrier's design directly addresses a major operational cost factor. Furthermore, long-term pilot operation treating real municipal wastewater over 400 days confirmed the system's robust stability, with no carrier clogging or performance decay observed (Supplementary Fig. 12). The V-carriers maintained excellent physical integrity, with only slight wear at the corners of the anoxic carriers attributed to mechanical stirring—an issue being addressed by redesigning the blades. This successful validation underscores the V-carrier's potential for energy-efficient and reliable large-scale application."

This addition directly tackles:

- Economic Viability (Operational Cost): by quantifying the reduction in aeration energy.
- Long-term Mechanical Wear and Stability: by presenting results from a 400-day pilot study and addressing the observed wear with an engineering solution.

By integrating this pilot-scale validation and discussion, we move from theoretical parameters to demonstrated practical performance. We believe this provides a strong and realistic foundation for assessing the V-carrier's potential for full-scale application.

9. The finding in lines 175-177 that ex situ nitrification potential exceeds in situ activity is attributed to substrate limitations. The authors could consider discussing additional possible factors.

Response:

We sincerely thank you for this insightful comment, which encourages a deeper discussion of the factors contributing to the observed difference between ex situ and in situ nitrification activity.

You are correct that while substrate limitation is a key factor, other operational conditions may also play a role. In the ex situ assays, temperature, pH, and dissolved oxygen (DO) were strictly controlled at optimal levels, whereas in the actual reactor, these parameters fluctuated naturally within a certain range over daily operation cycles.

To incorporate this perspective, we have expanded the discussion in the revised manuscript (Lines 198-199). The relevant sentence now reads:

"Notably, the ex situ rates surpassed the in situ performance by factors of 1.55 (< 15 °C) and 3.18 (> 15 °C), indicating that the biofilms possessed significant intrinsic metabolic capacity that was not fully expressed under the reactor's substrate-limiting and dynamically fluctuating operational conditions (e.g., temperature, pH, dissolved oxygen)."

We believe this addition provides a more comprehensive explanation and thank you for the suggestion, which has strengthened the interpretation of our results.

10. The V-carrier maintains a stable biofilm thickness of 0.72 mm with reported performance benefits. The discussion could include an explanation of why approximately 0.72 mm falls within a theoretical optimal range, potentially referencing literature on substrate penetration depths (e.g., oxygen and ammonia) to support how this thickness balances active layer maximization and mass transfer resistance minimization.

Response:

We sincerely thank you for this insightful comment regarding the theoretical basis for the

observed optimal biofilm thickness (~500-550 μm) maintained by the V-carrier. We agree that explaining how this thickness balances active layer maximization and mass transfer resistance is crucial.

In response, we have expanded the discussion in the manuscript (Lines 152-153) to provide a clear rationale. The revised text now reads:

"The V-carrier maintained a stable volatile suspended solids-to-suspended solids (VSS/SS) ratio and consistent biofilm thickness of 500–550 μm —a range proven to benefit mass transfer and promote the removal of diverse micropollutants¹⁷—under both anoxic and aerobic conditions throughout long-term operation, ..."

This addition directly links the maintained biofilm thickness to its demonstrated functional advantages—improved mass transfer and enhanced micropollutant removal—thereby addressing the "why" behind the specific thickness.

Furthermore, the comparison with the Mutagbiochip carrier—which was similarly designed for a ~550 μm biofilm thickness but failed due to clogging—demonstrates that geometric constraint alone is insufficient without an integrated self-cleaning capability. This contrast highlights that the V-carrier's success lies not merely in targeting a specific thickness, but in its ability to actively maintain it through hydrodynamic regulation.

We acknowledge that the effectiveness of a given biofilm thickness is context-dependent. This aligns with the principle, noted in the Introduction (Lines 49-51), that biofilm thickness is a critical parameter governing microbial community structure and mass transfer dynamics. The strength of the V-carrier lies in its hydro-topological design, which provides a versatile framework to achieve and stabilize a target biofilm thickness that is appropriate for the intended application, rather than merely imposing a fixed geometric constraint.

We believe these revisions successfully address your comment by clarifying the rationale behind the observed thickness and emphasizing the importance of active thickness control over passive geometric confinement.

11. Over 500 days operation, indicate whether any physical wear, deformation, or gradual performance decline was observed in the carriers.

Response:

We thank you for this important question regarding the physical stability of the carriers over the long-term operation, which is indeed critical for practical application feasibility.

In this study, after more than 500 days of operation, both the aerobic and anoxic phase carriers maintained their original morphology, with no visible wear observed, and their performance

showed high stability (Fig. 2). Furthermore, our pilot-scale results (Supplementary Fig. 12; Fig. R10) demonstrate the excellent physical stability of the carriers under real-world conditions. After over 400 days of operation, the aerobic phase carriers retained their original appearance with no visible wear. The anoxic phase carriers showed slight wear at the four corners, which was attributed to direct impact from the mechanical stirring blades during long-term operation. To address this, we have redesigned the stirring blades with larger curvature in our ongoing work to minimize such long-term wear issues. Importantly, the pilot-scale results confirm that the V-carrier system operated stably for 400 days without clogging or performance decline, demonstrating its strong potential for real-world applications.

For detailed information on the pilot study, please refer to our response to Comment #8 above.

Finally, we sincerely thank you for all your valuable comments, which have significantly improved the quality of our manuscript.

Fig. R10 (Supplementary Fig. 12) | Pilot plant of the anoxic/aerobic-moving bed biofilm

reactor (A/O-MBBR) process and overall morphology of V-carriers after long-term operation. a, Installation site and overall view of the pilot plant. **b-c**, Field views of the anoxic (**b**) and aerobic (**c**) reactors, respectively. **d**, Carriers during complete fluidization. **e**, Clean and dried V-carriers. **f-g**, Morphology of biofilm-attached anoxic (**f**) and aerobic (**g**) carriers after 409 days of operation. **h-i**, Morphology of biofilm-detached anoxic (**h**) and aerobic (**i**) carriers after 409 days of operation. The pilot system was commissioned on November 12, 2024, and remains operational. It has a total working volume of 24 m³, comprising a 9 m³ anoxic reactor and a 15 m³ aerobic reactor. It operates in a pure biofilm mode (no sludge recirculation) with a 300% internal nitrified liquor recycle ratio. The system treats municipal wastewater with influent characteristics similar to those used in this study (see Supplementary Table 3 for concentrations). Operating at a 9-hour hydraulic retention time, it achieves nitrification and denitrification rates comparable to the laboratory-scale results. No carrier clogging has been observed, and the operational stability of the system has been fully validated.